# A cost efficient spatially balanced hierarchical sampling design for monitoring boreal birds incorporating access costs and habitat stratification

Steven L. Van Wilgenburg[1]*, C. Lisa Mahon[2,3], Greg Campbell[4], Logan McLeod[2], Margaret Campbell[2], Dean Evans[5], Wendy Easton[6], Charles M. Francis[7], Samuel Haché[5], Craig S. Machtans[2], Caitlin Mader[8], Rhiannon F. Pankratz[5], Rich Russell[7], Adam C. Smith[7], Peter Thomas[9], Judith D. Toms[3,8], Junior A. Tremblay[10]

1 Environment and Climate Change Canada, Saskatoon, SK, Canada, 2 Environment and Climate Change Canada, Whitehorse, YT, Canada, 3 Department of Biological Sciences, University of Alberta, Edmonton, AB, Canada, 4 Environment and Climate Change Canada, Mount Pearl, NL, Canada, 5 Environment and Climate Change Canada, Yellowknife, NT, Canada, 6 Environment and Climate Change Canada, Vancouver, BC, Canada, 7 Environment and Climate Change Canada, Ottawa, ON, Canada, 8 Environment and Climate Change Canada, Edmonton, AB, Canada, 9 Environment and Climate Change Canada, Sackville, NB, Canada, 10 Environment and Climate Change Canada, Quebec City, QC, Canada

* steven.vanwilgenburg@Canada.ca

**Data Availability Statement:** All relevant data are within the manuscript and its Supporting Information files.

## Abstract

Predicting and mitigating impacts of climate change and development within the boreal biome requires a sound understanding of factors influencing the abundance, distribution, and population dynamics of species inhabiting this vast biome. Unfortunately, the limited accessibility of the boreal biome has resulted in sparse and spatially biased sampling, and thus our understanding of boreal bird population dynamics is limited. To implement effective conservation of boreal birds, a cost-effective approach to sampling the boreal biome will be needed. Our objective was to devise a sampling scheme for monitoring boreal birds that would improve our ability to model species-habitat relationships and monitor changes in population size and distribution. A statistically rigorous design to achieve these objectives would have to be spatially balanced and hierarchically structured with respect to ecozones, ecoregions and political jurisdictions. Therefore, we developed a multi-stage hierarchically structured sampling design known as the Boreal Optimal Sampling Strategy (BOSS) that included cost constraints, habitat stratification, and optimization to provide a cost-effective alternative to other common monitoring designs. Our design provided similar habitat and spatial representation to habitat stratification and equal-probability spatially balanced designs, respectively. Not only was our design able to achieve the desired habitat representation and spatial balance necessary to meet our objectives, it was also significantly less expensive (1.3−2.6 times less) than the alternative designs we considered. To further balance trade-offs between cost and representativeness prior to field implementation, we ran multiple iterations of the BOSS design and selected the one which minimized predicted costs while maximizing a multi-criteria evaluation of representativeness. Field implementation of the design in three vastly different regions over three field seasons showed that the

**Funding:** This work was supported with funding from Environment and Climate Change Canada.

**Competing interests:** The authors have declared that no competing interests exist.

approach can be implemented in a wide variety of logistical scenarios and ecological conditions. We provide worked examples and scripts to allow our approach to be implemented or adapted elsewhere. We also provide recommendations for possible future refinements to our approach, but recommend that our design now be implemented to provide unbiased information to assess the status of boreal birds and inform conservation and management actions.

## Introduction

Tackling ongoing [1–3] and projected [4,5] global biodiversity losses will require difficult decisions about resource allocation and the need to consider where, when, and how to focus conservation efforts. Species conservation is often most successful and least expensive when implemented early, before species require dedicated recovery efforts [6]. Early identification of species declines should also improve the likelihood of successful conservation. In addition, developing effective conservation strategies requires identification of the locations and factors contributing to species' declines [7]. Sparse or biased data may inaccurately identify which species require conservation actions or misidentify key drivers, resulting in misdirected or ineffective conservation actions [8]. Thus, well-designed ecological monitoring is necessary to prioritize conservation activities effectively [9–11].

Even comparatively well-monitored taxa such as terrestrial birds have significant gaps in species coverage that hinder effective status and trend assessments and associated conservation actions [12]. In North America, many species of terrestrial birds are primarily monitored using the North American Breeding Bird Survey (*hereafter* BBS; [12]). However, this survey, which is based on roadside surveys and mainly run by volunteers, has very limited coverage in remote northern locations such as the boreal forest [13,14]. Furthermore, much of the coverage for species that breed in the boreal forest is limited to the southern edges of their range, where population change may be quite different from elsewhere. The lack of data from most of the boreal forest may present significant risks to conservation given expanding resource development [15–17] and projected climate change impacts on boreal birds [18].

Monitoring data from northern ecoregions are needed to inform conservation actions (e.g., prioritize selection for protected areas), contribute to management decisions (e.g., adaptive management), detect range shifts, assess threats to species (e.g., species response to developments), and meet legislative requirements. In Canada, monitoring data are required to support conservation under the Migratory Birds Convention Act (1994, c. 22), which includes informing listing decisions under Canada's Species at Risk Act (*hereafter* SARA) [19], and environmental impact assessments required by the Impact Assessment Act [20] or by provinces and territories. Key criteria used by the Committee on the Status of Endangered Wildlife in Canada (COSEWIC) to recommend listing a species under SARA include rate of decline in the total number of mature individuals, population size, and extent of species occurrence [21]. In addition, habitat- and region-specific estimates of species' abundance or density are required as baseline data in environmental impact assessments [20]. To support these diverse requirements, multi-species bird monitoring should be capable of accurately capturing not only trends in species' population sizes, as well as spatial and temporal variation in species' status and distribution.

Given the breadth of these objectives and the large and remote areas to be monitored, a cost-effective sampling approach is required. As most of these locations have limited road

networks [22] with few human settlements, there is little potential for volunteer surveys [23]. Similarly, a simple, design-based sampling strategy (e.g., a randomized design in which sample units are selected with the same probabilities) is logistically and financially impractical [9,24]. Spatially balanced sampling offers an attractive alternative as it tends to be more efficient, providing more precise estimates than alternative designs [25–28]. Despite the advantages offered by spatial sampling designs, they have not been more widely implemented in large-scale (e.g., biome- or continent-wide) studies, perhaps because they force sampling into locations that can be costly to access. Instead, we propose a design-based sampling scheme that integrates multiple approaches from the statistical sampling-theory literature, including spatially balanced sampling [25,26], stratified sampling [29–32], cost-constrained sampling [31] and optimization [33,34]. The integration of these approaches facilitates simultaneous maximization of both spatial balance (i.e., evenness of the spatial coverage) and habitat representation, while minimizing program costs.

Here, we describe a cost-effective hierarchically structured unequal-probability sampling design known as the Boreal Optimal Sampling Strategy (BOSS). Specifically, our objectives are to: (1) outline the development of the BOSS sampling design; (2) describe BOSS design implementation in three ecologically distinct regions in Canada (Newfoundland and Labrador, Saskatchewan, Yukon Territory); (3) evaluate how the BOSS design meets spatial balance, habitat representation, and cost objectives in these three distinct regions; and (4) evaluate implementation costs in field trials. We compared our design against three competing spatially balanced sampling designs: a purely spatial (equal-probability) design, a habitat stratification design, and a design that minimized access costs. We hypothesized that the BOSS design should provide the second best habitat representation after the habitat stratification design since we incorporated habitat in the inclusion probabilities. We also hypothesized that our BOSS design should rank as the second-least expensive alternative after the cost design owing to incorporation of costs within the inclusion probabilities.

## Materials and methods

### Sampling frame

We defined the target population as all adult individuals of terrestrial bird populations occupying an ecoregion, province/territory (*hereafter* jurisdiction), and Bird Conservation Region (BCR) within the boreal region of Canada [35] during the breeding season for a particular year. We defined the Primary Sampling Unit (PSU) as a 5 km diameter hexagonal grid cell (i.e., circumscribed by a circle with a radius of 2500m). We generated a hexagonal grid of PSUs covering the entire sampling frame: the boreal region of Canada, plus a 100 km northern buffer (Fig 1). We used hexagonal geometry because it introduces fewer edge effects and performs well in nearest neighbour analyses since the distances between centroids are the same in all directions [36]. We included a northern buffer to allow the design to capture possible breeding range shifts associated with climatic changes in the north [37]. Each PSU was assigned a unique number/identifier. With the exclusion of PSUs that have cumulative inclusion probabilities of zero (e.g., exclusively open water or having too little land to fit secondary sampling locations), all PSUs in the sampling frame were available to be sampled. As a result, the BOSS sampling design can make valid inferences to entire populations of species inhabiting the boreal region [38].

### Stratification

We developed a hierarchical stratification scheme to estimate population variables at multiple spatial scales. At the first level, we stratified the sampling frame using the intersection of BCRs

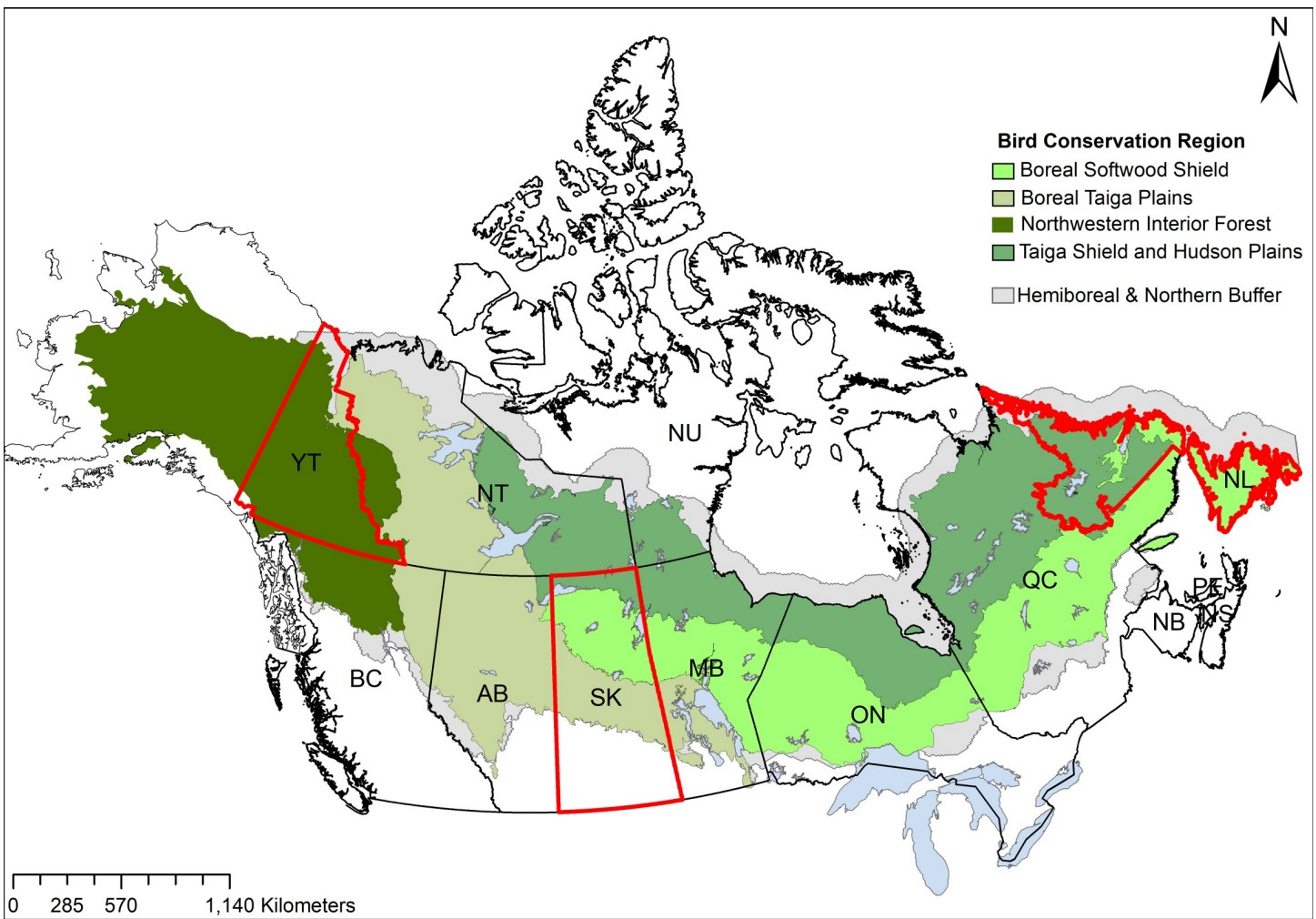

**Fig 1. Spatial extent of the sampling frame of the Boreal Optimal Sampling Strategy (BOSS) design.** The green region represents the boundary of the boreal region as defined in [35] with a 100 km northern buffer within the 4-Northwestern Interior Forest, 6-Boreal Taiga Plains, 7-Taiga Shield and Hudson Plains, and 8-Boreal Softwood Shield Bird Conservation Regions. Reprinted from [35] under a CC BY license, with permission from NRC Research Press, original copyright 2009.

and jurisdictions (e.g., area of Yukon Territory within BCR 4-Northwestern Interior Forest). At the second level, we further stratified each BCR by jurisdiction and ecoregion (e.g., the area of Selwyn Mountains ecoregion within Yukon Territory within BCR 4), where ecoregions are a subdivision of an ecozone characterized by similarity in surface forms, flora, fauna, hydrology, soil, and macro climate [39]. Regional monitoring programs as part of the BOSS design are defined by the area of all boreal ecoregions falling within a given jurisdiction. We set target sample sizes for the number of PSUs to select within an ecoregion using the process outlined below (see sample size allocation).

**Sample size allocation.** From sampling theory, a stratified sample is optimal when it provides the greatest precision for the lowest cost [40]. Optimal stratified sampling accounts for both the size of the stratum (e.g., land area for geographic regions), and the variance within the stratum [40]. By placing greater sampling effort in strata that are larger and more variable, the overall precision of the estimate(s) is optimized [29,31]. Here, we calculated the sample size of PSUs (n) for a given jurisdiction using the following equation to allocate sampling effort

between all (*k*) ecoregions within the jurisdiction:

$$n_e = N * \frac{Area_e * \sigma_e * r_e}{\sum_{e_{1...k}} Area_e * \sigma_e * r_e}$$ (1)

Where $n_e$ is the target sample size for a given ecoregion within a given jurisdiction, *N* is the total target sample size to sample within the jurisdiction, $Area_e$ is the area of ecoregion *e*, $\sigma_e$ is a measure of observed or predicted variation within ecoregion *e* for the variable(s) of interest, and $r_e$ is the described species richness of the bird community in ecoregion *e* (i.e., the number of all bird species with ranges overlapping the ecoregion derived from overlays of species range maps with the ecodistricts of Canada [39] by GeoInsight Corporation [41]. Since our goal is to monitor multiple species, employing a univariate measure of variance would provide an inadequate representation of variance [31]. We included species richness as a weighting factor on within-stratum variation ($\sigma_e$), to increase sampling in ecoregions where more species occur and improve (1) the likelihood of adequately monitoring a greater number of species, and (2) the ability to monitor and predict species distributions.

Many of the ecoregions in our sampling frame have little or no historic sampling from which we could derive estimates of spatial or temporal variance in the bird community. As an alternative, we used proxy variables that correspond with the spatial and/or temporal variance in bird abundance within an ecoregion to set relative PSU sample size targets for the ecoregion (i.e., as proxies for $\sigma_e$). We used these proxy variables [31] to calculate a measure of multivariate dispersion (variance estimate) for each ecoregion following [42,43], representing anticipated spatial and temporal variation in bird abundance and distribution. We used several well-described links between species abundance and distribution and spatial and temporal climatic variation [44], vegetation [44,45] and variation in forest age [45,46]. Specifically, we used the following proxy variables: (1) variance of mean temperatures for May-July across years (1981–2014); (2) variance of total precipitation for May-July across years (1981–2014); (3) standard deviation of annual burn rates (% area burned by wildfire per year [47]) between 1980 and 2015 within an ecodistrict; (4) richness of land cover classes [48] within an ecodistrict; (5) standard deviation of elevation; and (6) percent area of open water since riparian edges significantly increase avian species richness [49]. We calculated variance of mean May-July temperatures and variance of total precipitation from monthly gridded (0.5 x 0.5 degrees) values from the climate datasets of Harris et al. [50]. We calculated the between year standard deviation of burn rates from data obtained from [47]. Since burn rates from [47] were calculated in sample units of 10,000 km$^2$ hexagons, we first calculated the between-year standard deviation of burn rates within hexagons, and derived the mean of those values for ecodistricts. Land cover of Canada 2010 [48] is a 250 m resolution land cover layer of 15 land cover classes from 13,350 Landsat-7 satellite images taken between 2009 and 2011. Ecodistricts (subdivisions of ecoregions characterized by distinctive landforms, relief, geology, soils, water bodies, flora and fauna [39]) were treated as sampling units to allow us to calculate dispersion within ecoregions. Proxy variable data are provided in S1 Data.

## Sampling design

We developed a two-stage sampling design in which the first stage of the design involved selection of PSUs from ecoregions within the sampling frame and the second stage involved selection of Secondary Sampling Units (SSU) within the selected PSUs (Fig 2). The SSUs represent

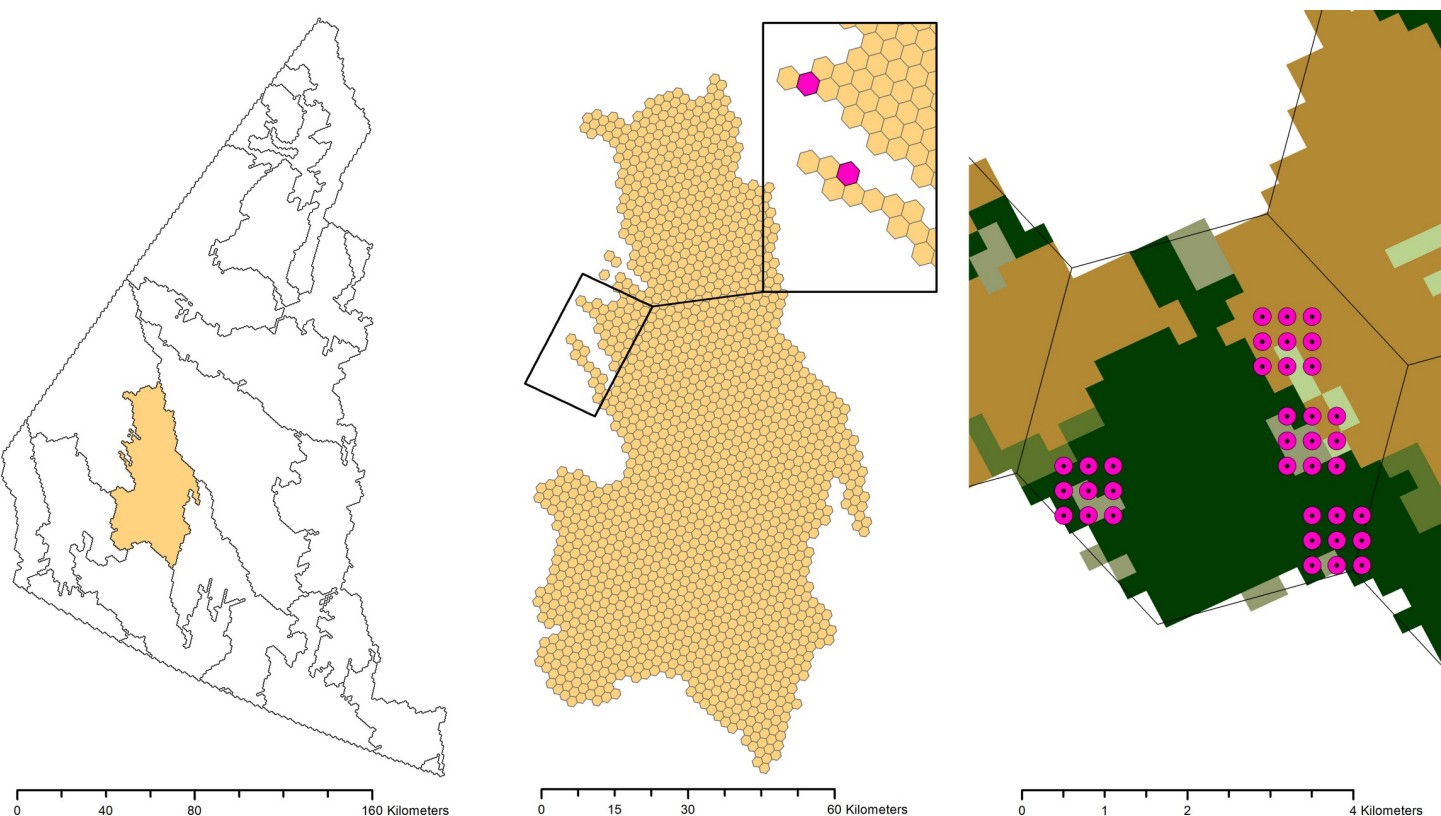

**Fig 2. The two-stage sampling design of the Boreal Optimal Sampling Strategy (BOSS) design.** This figure illustrates the selection of Primary Sampling Units (PSU, 5 km diameter hexagonal grid cell) from ecoregions within the sampling frame and selection of Secondary Sampling Units (SSU, point count survey locations) within the selected PSUs. Reprinted from [35] under a CC BY license, with permission from NRC Research Press, original copyright 2009.

on-the-ground locations where point count surveys [51] will be conducted using trained observers and/or autonomous recording units (*hereafter* ARUs) [52].

**Selection of primary sampling units.** To select PSUs within each jurisdiction (Stage 1), we first developed an approach to combine the following three design components: access costs, habitat representation, and spatial balance. We followed the steps outlined below in each of our three test jurisdictions (Newfoundland and Labrador, Saskatchewan, and Yukon Territory) to integrate all components of our sampling design across our sampling frame (Fig 3). First, we created separate cost surfaces for logistically feasible access methods within each of our jurisdictions. Specifically, we developed spatially explicit models predicting the cost to access a given PSU assuming access from roads (trucks), helicopters, float planes, canoe, motorized boat, snowmobile and/or all-terrain vehicles as appropriate for a given jurisdiction (see examples in S2 Data). The resulting cost models were represented as pixel-based (250 x 250m) estimates of cost, from which we subsequently calculated the average cost of access for all pixels within a PSU separately for each access type (e.g., trucks, helicopters, float planes). We assumed that the lowest-cost access method will generally be used to access a given PSU and calculated the minimum cost across all access cost surfaces. We subsequently used this value as the estimated access cost for each PSU.

Second, we calculated inclusion probabilities based on land cover classes (hereafter habitat classes) from the North American Land Change Monitoring System [48]. This land cover product consisted of nineteen land cover classes in total, of which only fifteen occurred in our sampling frame. We calculated habitat-based inclusion probabilities in an

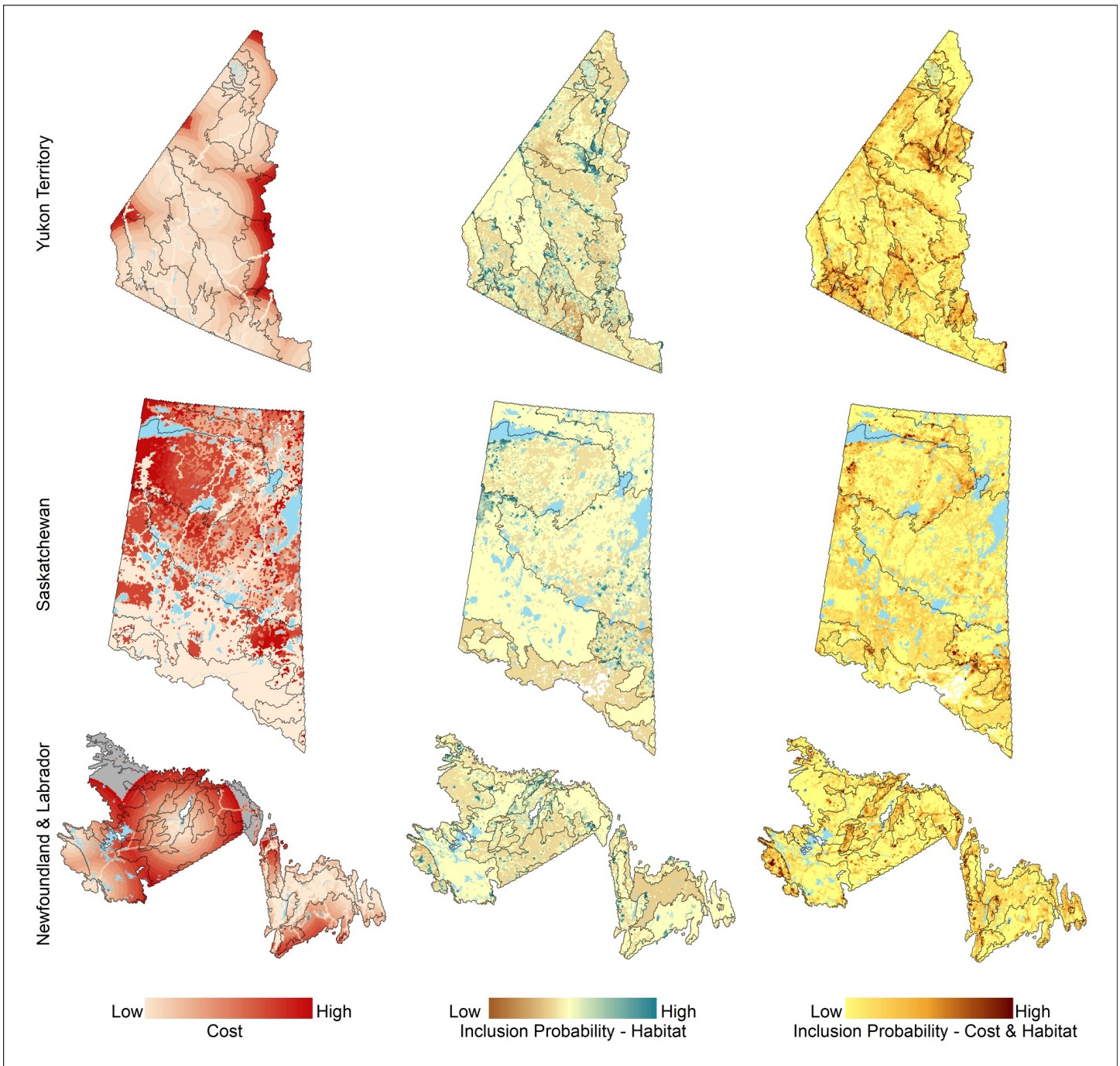

**Fig 3.** Components of the Boreal Optimal Sampling Strategy (BOSS) design depicted for Newfoundland and Labrador, Saskatchewan, and Yukon Territory: (a) access cost; (b) weighted habitat inclusion probability; (c) cost and weighted habitat inclusion probability. Reprinted from [35] under a CC BY license, with permission from NRC Research Press, original copyright 2009.

attempt to achieve balanced representation of habitat classes within an ecoregion. Inclusion probabilities for habitat classes used a weighted sampling approach to balance sampling across common (high occurrence/high proportion) and uncommon (low occurrence/low proportion) habitats within an ecoregion. Weighted habitat inclusion

probabilities were calculated as follows:

$$p_{habitatpie} = \frac{\frac{1}{of\ Habitat\ Classes\ in\ ecoregion_e}}{Area_{ie}} \tag{2}$$

where the inclusion probability for a given pixel $p$ of habitat class $i$ within a given ecoregion ($p_{habitatpie}$) was a function of the inverse of the number of habitat classes within the ecoregion divided by the area of habitat $i$ within ecoregion $e$. Our weighted habitat inclusion probability thus results in the cumulative probability of sampling a given habitat within an ecoregion being equal among all habitat classes despite differences in habitat area, to help ensure that we can draw inferences for rare habitat types.

Third, we combined access costs with weighted habitat inclusion probabilities to derive our final summed value for inclusion probabilities for each PSU as follows:

$$I_{SU_k} = \frac{\sum p_{habitatpie_{1...j}} * \frac{1}{\sqrt{cost_k}}}{\sum_{1...k}\left(p_{habitatpie_{1...j}} * \frac{1}{\sqrt{cost_k}}\right)} \tag{3}$$

Where $I_{SUk}$ = the sum of all inclusion probabilities within a given primary sample unit (PSU) 1 through $k$, $P_{habitatpie}$ = area-weighted selection probabilities for habitat class $i$ for pixels ($p$) 1 through $j$ in ecoregion $e$ per Eq 2, and cost for a given PSU was the minimum access cost for sample unit $k$ (as described above). Cost was included as the inverse square root of costs to preferentially (all else being equal) favour sampling at PSUs that are less expensive to sample in accordance with optimal allocation theory [53]. Across all sample units within the given study area, the inclusion probabilities sum to unity based on Eq 3.

Fourth, we used unequal-probability sampling based on the calculated inclusion probabilities ($P_{SUk}$) to draw a spatially balanced sample within each ecoregion using Generalized Random Tessellation Stratified sampling (GRTS; [25]). In brief, the GRTS approach maps two-dimensional space into a single dimension (that maintains the spatial properties of the original space) and takes a systematic sample along that representation, yielding a sample that is neither over- nor under-represented across space. For a detailed description, we refer the reader to [25] and [30]. The random draw to select PSUs within ecoregions in each jurisdiction uses PSU sample size targets for each ecoregion (where ecoregion boundary was clipped to the jurisdiction boundary). We included a 20% oversample of PSUs for each ecoregion to allow substitution of inaccessible or unsafe PSUs (e.g., cliffs or other features not visible in the available input layers). All PSUs were labeled with their draw order and oversample PSUs were additionally labeled as oversamples. Should a PSU be inaccessible or unsafe to survey, the oversample and draw order can be used to select a replacement PSU that retains the spatial balance of the sample [25].

Finally, we derived an approximately optimal draw of PSUs using a combination of repeated random sampling and multi-criteria evaluation. Specifically, we propose that multiple random unequal-probability GRTS samples can be obtained for which the predicted sampling costs and metrics of habitat representation and spatial balance can be derived. The iteration resulting in the highest combined representativeness of habitat sampling combined with spatial balance for the lowest predicted sampling cost should provide an approximately optimal design. As a metric of habitat representation, we calculated the sum of squared differences (SSD) between the area of each habitat class within all PSUs in a randomized draw versus the proportional area of each habitat class. As a metric of spatial balance, we use Pielou evenness (PE in Eq 4 below) to represent the even spread of PSUs across the sample frame [30]. Possible values of PE range from 0–1. In order to evaluate which draw represents the best trade-off of

cost versus combined spatial and habitat representation across multiple (1...$j$) iterations, we propose evaluating which iteration minimizes predicted costs while maximizing the weighted sum [34] of the two metrics:

$$((1 - (SSD_i - \min(SSD_{i...j}))) \bullet w_h) + (w_s \bullet PE_i) \tag{4}$$

Where $SSD$ for sample draw $i$ is relativized to the minimum $SSD$ across draws 1 through $j$, by subtracting the resulting value from 1 so that the metric scales from 0–1 and thus a value of 1 would represent idealized representation of habitat composition within the draw. We applied equal weights to both the habitat ($w_h$) and spatial balance ($w_s$) components as we did not want either component to be more heavily valued than the other in our final design (i.e., $w_h = w_s = 0.5$). However, this approach can be generalized by allowing weights to vary between habitat representation and spatial representation if there is a desire to emphasize habitat representation (e.g., for environmental impact assessments) or spatial representation (e.g., for distribution modeling).

**Selection of secondary sampling units.** To select secondary sample units (SSU) within each PSU in each jurisdiction (Stage 2), we first created a uniform (systematic) grid of potential secondary sample unit locations at 300 m spacing over the extent of each ecoregion. Each SSU was separated by 300 m since the effective detection radius for most boreal songbirds is less than 150 m [51] and therefore double counting should be infrequent for most species. We used the grid of SSUs to query the habitat-based inclusion probabilities. Within each PSU, we selected a stratified random sample of four SSUs using inclusion probabilities as calculated in Eq 2. We used the four SSUs selected during this draw as plot centroids and the eight SSUs surrounding the plot centroid to construct plots of nine SSUs for sampling (see Fig 2). We set a sampling target of two plots of nine SSUs per PSU so that a team of two staff can complete one PSU per day in most circumstances. Furthermore, having two survey plots within a PSU allows staff to have a partner(s) to provide support in an emergency situation. Our PSU size of 5 km diameter was specifically selected to ensure field staff would be working within a reasonable walking or flight distance of a partner if assistance or emergency response is required. We drew a sample of four plots of SSUs per PSU to create an oversample in case of inaccessibility (e.g., due to large water barrier or cliff) or safety concerns (e.g., hazardous terrain or aggressive wildlife).

**Data collection.** We tested the BOSS design within three jurisdictions across the boreal biome of Canada (Newfoundland and Labrador [NL], Saskatchewan [SK], and Yukon Territory [YT]) during 2017, 2018, and 2019. Newfoundland and Labrador is located at the eastern edge of the boreal region within BCR 7-Taiga Shield and Hudson Plains and BCR 8-Boreal Softwood Shield (Fig 1). Saskatchewan is located in the center within BCR 6-Boreal Taiga Plains, BCR 8, and BCR 7 (see Fig 1); while the Yukon Territory is located at the northwestern edge within BCR 4-Northwestern Interior Forest, BCR 6, and BCR 3-Arctic Plains and Mountains. Jurisdictions were also ecologically distinct, differing in their geology, topography, vegetation, terrestrial bird diversity, and climate as indicated by the number of ecozones (NL: 2; SK: 3; YT: 3), ecoregions (NL: 23; SK: 8; YT: 22), and elevation range (NL: 0–1,652 m; SK: 204 −823 m; YT: 0–5,900 m). Differences in the location of mountain ranges, rivers and lakes, human settlements, and primary and secondary road networks within each jurisdiction present distinct logistical challenges and therefore useful comparative test cases. To conduct point count surveys at SSUs within selected PSUs, we used 10-minute point counts with trained observers following standard protocols recommended by Ralph et al. [54] and Matsuoka et al. [51] and conducted during suitable weather conditions from 30 minutes prior to sunrise until 4–5 hours after sunrise, from the last week of May to the first week of July. Where we

conducted surveys using ARUs, we analysed 10-minute recordings from [52] to match the count duration conducted by human point counts. ARUs were deployed on a variety of different schedules and using multiple access methods. In some cases, ARUs were deployed in February by snowmobile, and were retrieved in the summer. Wherever possible, we attempted to get recordings conducted over $\geq$ 4 mornings with good survey conditions; however, ARUs carried on canoe trips were deployed for as little as a single night. We programmed ARUs to record a minimum of six 10-minute intervals over the course of a morning during the same time-period as point counts, as well as additional times for other objectives. An analysis of the relative merits of different programming schedules and a comparison of field observers with ARUs is beyond the scope of this paper.

We sampled 295 PSUs selected using the BOSS design between 2017–2019 in Newfoundland and Labrador (n = 69), Saskatchewan (n = 103) and Yukon Territory (n = 123). Three PSUs in Saskatchewan had to be replaced with PSUs from the oversample owing to access and safety issues associated with two open-pit uranium mines and one on a bombing range within Department of National Defence lands. No oversamples were required in either Newfoundland and Labrador or Yukon Territory. In Newfoundland and Labrador, we sampled 5 out of 9 ecoregions on the island of Newfoundland, with all ecoregions being in BCR 8. PSUs were sampled in the Avalon Forest ecoregion (n = 13), Central Newfoundland (n = 1), Maritime Barrens (n = 8), South Avalon-Burin Oceanic Barrens (n = 23), and Southwestern Newfoundland (n = 24). We sampled seven out of eight ecoregions within Saskatchewan: Selwyn Lake Upland (BCR 7, n = 12), Tazin Lake Upland (BCR 7, n = 9), Athabasca Plain (BCR 8, n = 19), Churchill River Upland (BCR 8, n = 22), Mid-boreal Uplands (BCR 6, n = 28), Mid-boreal Lowland (BCR 6, n = 5), and the Boreal Transition (BCR 6, n = 8). We sampled ten out of 22 ecoregions within Yukon Territory, sampling in the British-Richardson Mountains (BCR 3, n = 1), Old Crow Basin (BCR 4, n = 9), Old Crow Flats (BCR 4, n = 2), Eagle Plains (BCR 4, n = 3), North Ogilvie Mountains (BCR 4, n = 5), Mackenzie Mountains (BCR 4, n = 8), Klondike Plateau (BCR 4, n = 72), Yukon Plateau Central (BCR 4, n = 10), Yukon Plateau North (BCR 4, n = 10), and Ruby Ranges (BCR 4, n = 3).

**Ethics statement.** We obtained the required permits to conduct avian monitoring to test the BOSS design from provinces/territories, Provincial and Territorial Parks, Parks Canada Agency, and Yukon First Nations with Traditional Territories and Settlement Areas. Field observers spend <15 minutes at SSU locations during observer surveys and <30 minutes (time for set up and subsequent retrieval) at SSU locations during ARU surveys, resulting in minimal disturbance to breeding birds.

## Statistical analyses

Prior to analysis of our multivariate proxy data, we centered and standardized all variables to zero means and unit variance and then calculated Euclidean distances between all points (ecodistricts) in the dataset. We derived the multivariate estimate of dispersion by calculating mean distances to ecoregion centroids using the 'betadisper' function in the vegan package [55] in the R statistical computing environment [56]. The resulting estimates of dispersion for all ecodistricts within the boreal region, as per Brandt [35], are available in S3 Data.

In order to assess our BOSS design, we used the 'spsurvey package' [57] within the R statistical computing environment [56] to draw samples under four alternative spatially balanced sampling designs. Specifically, we drew samples under one equal-probability spatially balanced design and three unequal-probability designs: (a) a habitat stratification design, calculated as per Eq (2) above; (b) a cost design in which inclusion probabilities were based solely on access costs as per Eq (3); and (c) our BOSS design in which inclusion probabilities included both

habitat stratification and access costs as per Eq (4) above. For each scenario, we ran 100 iterations in which we used the GRTS algorithm [57] to draw a random sample of n = 400 PSUs from each of the three jurisdictions considered here. Within each iteration, our script calculated the access costs (CDN $), spatial balance (Pielou evenness [30,57]), and habitat representation as SSD (see above). All implementations are available within scripts in the (S2 and S6 Datas) along with geospatial data examples (S4, S5 and S7 Datas).

We used linear models to test for differences (for each jurisdiction separately) in each of our response variables (cost, SSD, and Pielou evenness) between the alternative designs described above (i.e., spatial, habitat, cost, and BOSS). Study design was included as a factor, and in each model we specified an *a priori* reference factor level to test our hypotheses. We specified the cost design as the reference factor level for models examining access costs, as we anticipated this design should have the lowest access costs on average. Similarly, we specified the habitat stratification design as the reference level for models examining variation in SSD as we predicted this design should have the lowest SSD.

All four designs we considered employed a GRTS algorithm to generate spatially balanced samples; however, the use of unequal inclusion probabilities based on spatially structured variables could alter the ability of the design to achieve spatial balance. We therefore tested for differences in spatial balance between our designs and we treated the spatial design (equal-probability spatial sampling design) as the reference factor level when comparing spatial balance between sampling designs. We calculated robust standard errors ('sandwich' estimators) to overcome problems with heteroscedastic errors using the 'lmtest' [58] and 'sandwich' [59] packages within the R statistical computing environment [56]. All data are available in the (S8 Data).

## Results

### Sample size allocation

Sample size allocation to geographic strata (ecoregions × jurisdiction) under the BOSS design resulted in a similar sample size on average as allocating samples based on stratum area, because data are centered along a 1:1 correspondence line (Fig 4). However, allocation to individual strata differed markedly, with the BOSS design suggesting significantly lower sample sizes for some strata than an area-based allocation, and allocating disproportionately larger sample sizes in other strata (Fig 4). For example, sampling proportional to area would allocate 151 PSUs to the New Québec Central Plateau ecoregion of Québec, compared to only 50 based on the BOSS allocation, because the stratum has lower than average dispersion (i.e., 0.36 vs average of 1.04). In contrast, 103 PSUs would be allocated to the Abitibi Plains ecoregion of Ontario under sampling proportional to stratum area, but 173 PSUs were allocated to the same stratum under the BOSS design owing to higher than average dispersion (i.e., 20.1 vs. average of 1.04). Several of the larger and more variable strata (sample sizes > 100) require more sampling under the BOSS design than predicted solely based on stratum area (Fig 4). Overall, the BOSS sample size allocation generally results in sampling intensities being greatest in the south and decreasing northward (Fig 5).

### Predicted sampling costs

Predicted costs to access 400 primary sample units varied by the jurisdiction and sampling design considered (Fig 6). Across all three jurisdictions, designs in which the inclusion probabilities were derived solely based on access costs (cost design) were the least expensive designs (Fig 6) and all the alternative designs were more expensive regardless of jurisdiction (Table 1). The BOSS design had the next lowest cost (Fig 6), with costs predicted to be between 1.02

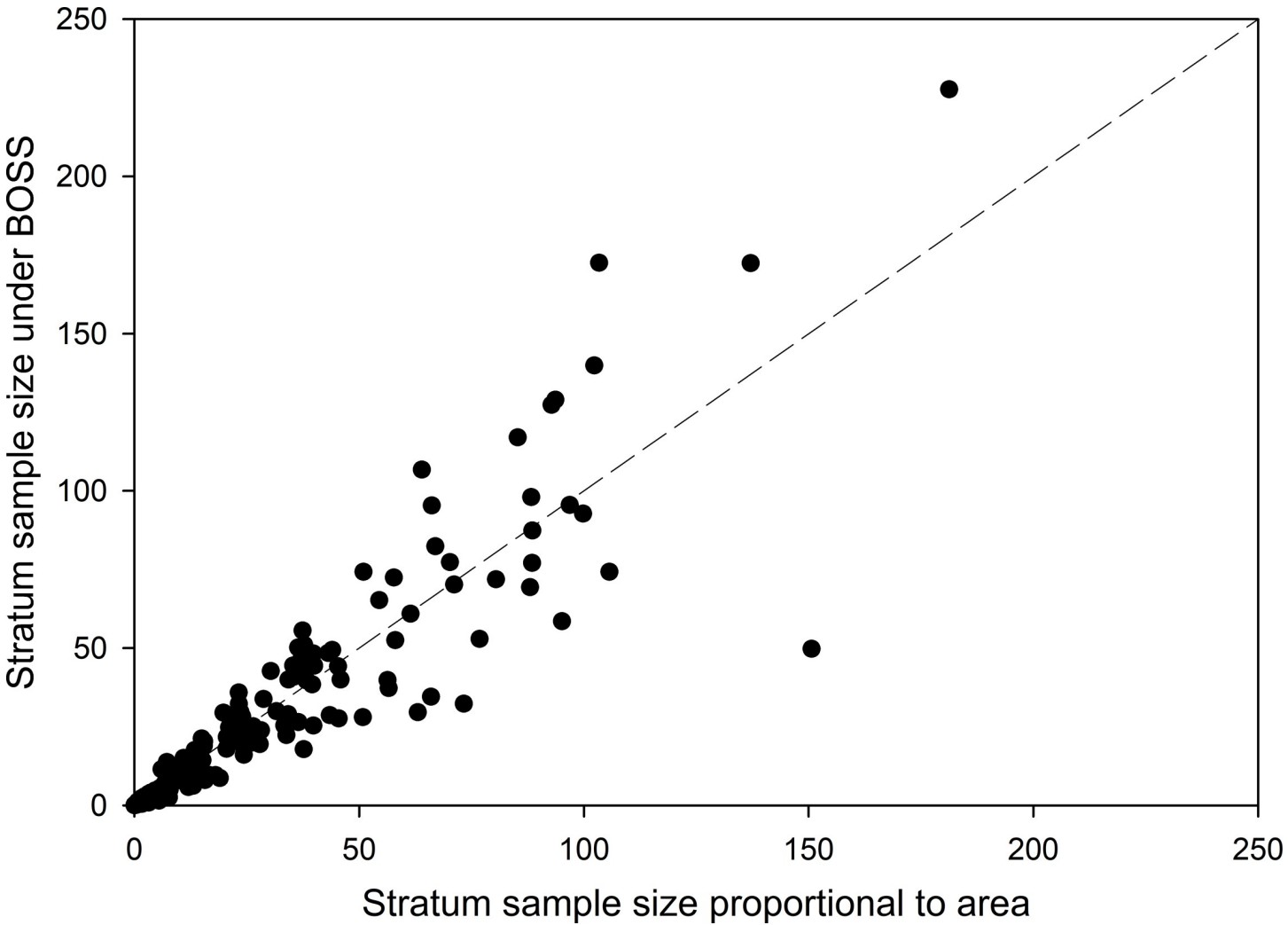

**Fig 4. Allocation of sampling effort between boreal ecoregions of Canada under the Boreal Optimal Sampling Strategy (BOSS; see Methods) versus traditional allocation based on Stratum areas to achieve a total sample size of 4980 primary sample units (i.e. 2% sample).** Dashed line indicates the 1:1 correspondence line.

(Yukon Territory) and 1.2 (Saskatchewan) times more expensive on average than the cost design for a given jurisdiction (Table 1). In both Newfoundland and Saskatchewan the equal-probability spatial design had the third-lowest cost followed by the habitat stratification design, whereas it was the most expensive design for Yukon Territory (Fig 6); it was predicted to be 1.4–2.6 times more expensive on average (~$362,000–562,000; Table 1) than the cost design (maximum of 1.5–3.0 times; Fig 6). The habitat stratification design was predicted to be 1.4–2.0 times more expensive on average (~$122,000–1,020,000; Table 1), or a maximum of 1.6–2.3 times more expensive across draws and jurisdictions (Fig 6).

### Habitat representation

As predicted, the habitat stratification design had the lowest sum of squared differences between the habitat areas represented within sample draws and the desired equal habitat representation (Fig 7). The BOSS design provided the second-most representative sampling of habitat classes on average across all three jurisdictions (Fig 7 and Table 2). In contrast, both the

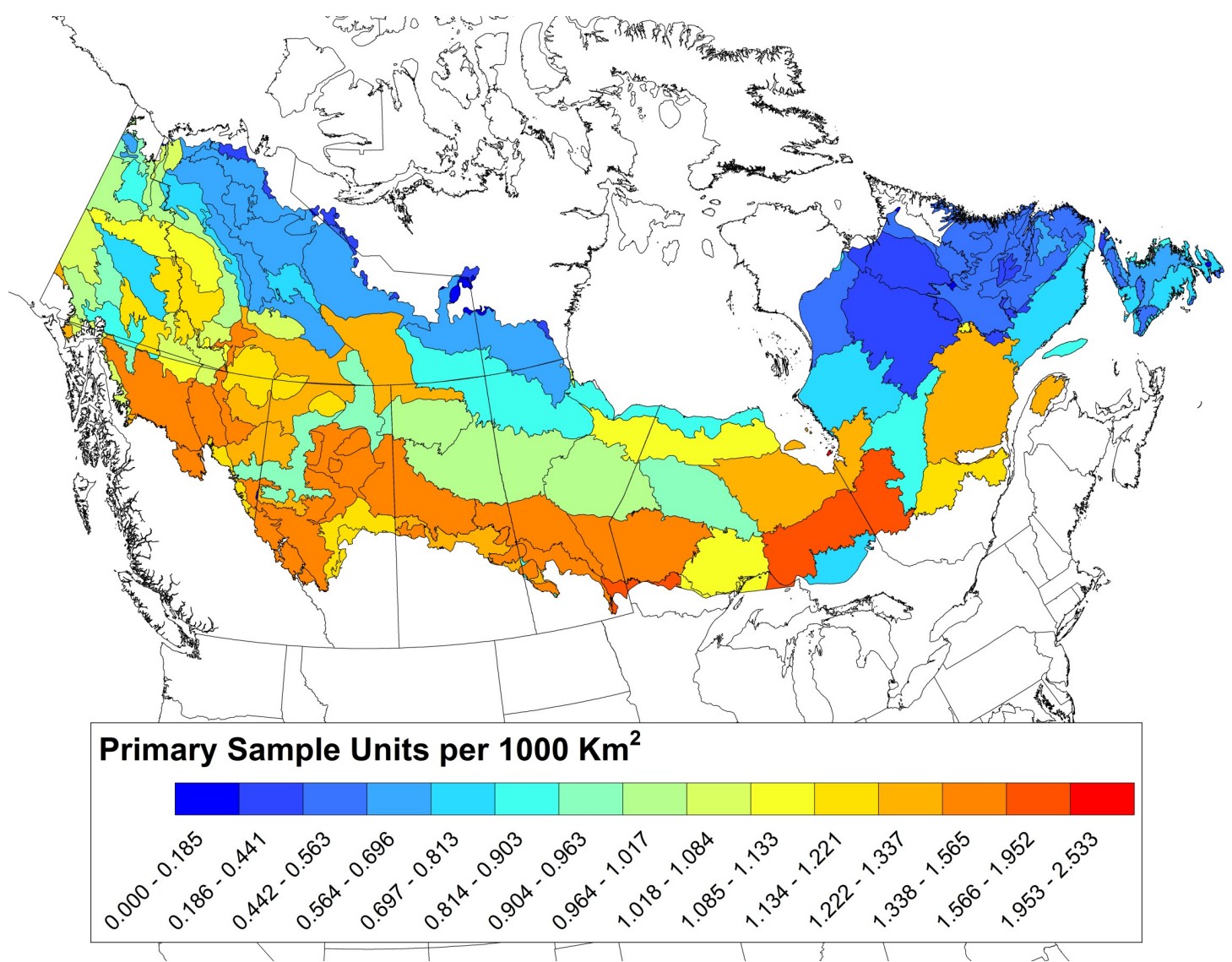

**Fig 5. Geographic distribution of sampling intensity (number of allocated primary sampling units per 1000 km²) across the boreal ecoregions of Canada under the Boreal Optimal Sampling Strategy (BOSS) design (see Methods).** Reprinted from [35] under a CC BY license, with permission from NRC Research Press, original copyright 2009.

equal-probability spatial design and the cost design (Fig 7) had substantially lower habitat representation relative to the habitat stratification design in all three jurisdictions (Table 2).

## Spatial representation

All sampling designs provided relatively balanced spatial sampling of all three jurisdictions (range of Pielou evenness across all designs: 0.96–1.01; Fig 8). Interestingly, the spatial balance of the cost design provided the closest approximation to the equal-probability spatial design (Fig 8) and with substantial overlap of the data distribution compared to the equal-probability spatial design (Table 3). The BOSS design showed small differences in spatial balance relative to a purely spatial design in both Newfoundland and Yukon Territory (Fig 8 and Table 3), but not Saskatchewan (Table 3). The habitat-stratification design had the largest overall differences

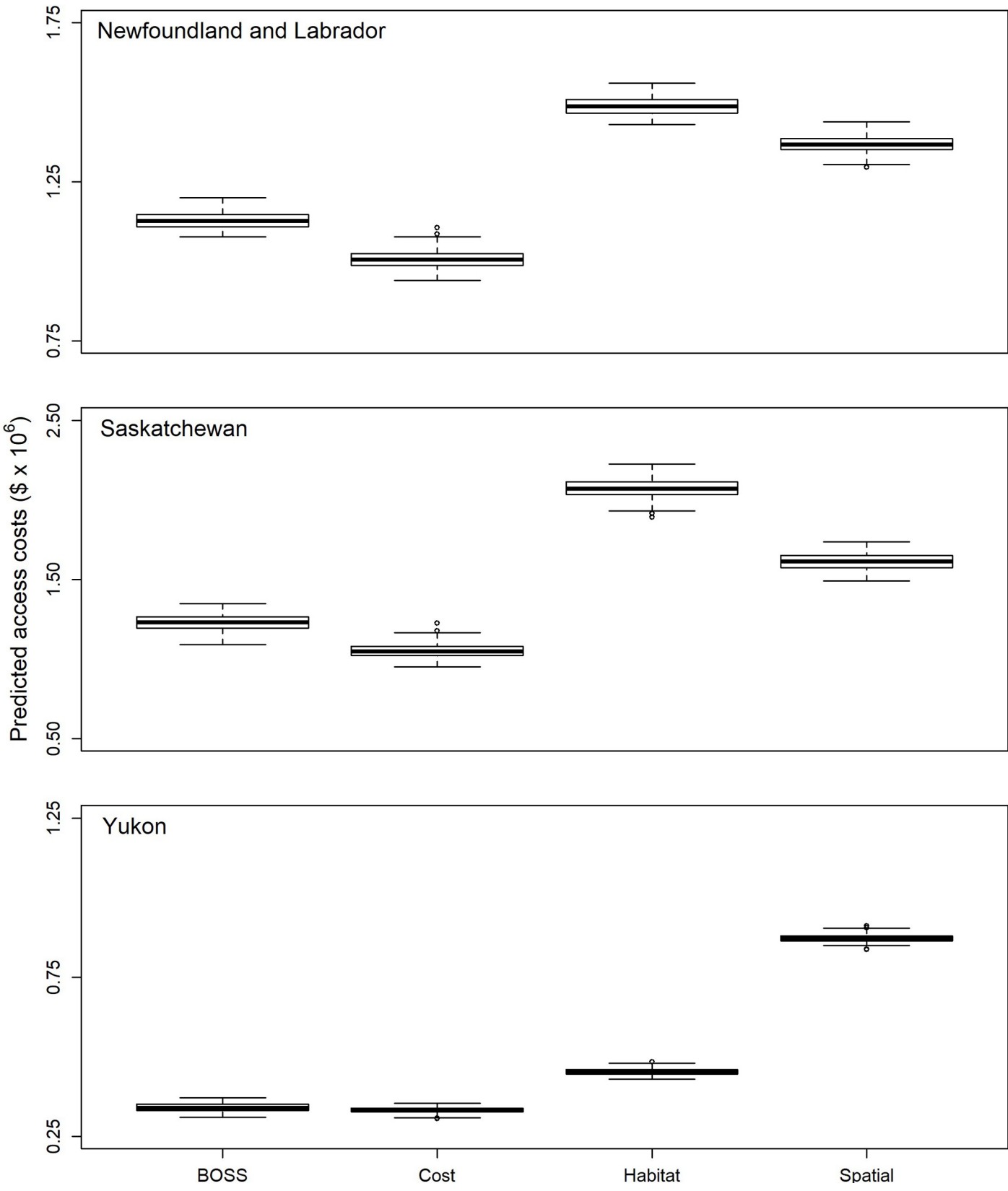

**Fig 6. Variation in predicted costs (in Canadian dollars x 10⁶) of accessing 400 primary sampling units in each of Newfoundland and Labrador, Saskatchewan and Yukon Territory.** Data are from 100 separate random draws of 400 primary sampling units in each jurisdiction. Note differing y-axis scales due to variation in overall costs of operating in each jurisdiction. Dark solid line represents the median, box indicates the inter-quartile range, whiskers are 1.5 times the inter-quartile range, and dots indicate extreme values.

in spatial balance relative to the equal-probability spatial design, with the largest difference occurring in Yukon Territory (Fig 8 and Table 3).

## Optimal design selection

Across 100 random sample draws, the BOSS design provided the best combination of cost savings and both spatial and habitat representation amongst the four designs considered. Regardless of jurisdiction, the BOSS design had costs competitive with the cost design (Figs 6 and 9), with no loss of habitat or spatial representation. For all three jurisdictions combined, selecting the draw that simultaneously maximized representation while minimizing cost (see red-filled squares in Fig 9) only results in an estimated savings of approximately $2,000 relative to the median of all BOSS sample draws. However, access costs for 400 PSUs varied substantially between draws (Newfoundland: $1,076,897–1,199,693; Saskatchewan: $1,089,811 −1,348,550; and Yukon Territory: $309,497−371,174) and thus relative to the most expensive draws for each jurisdiction, the combined savings could be up to an estimated $225,000 for these three jurisdictions when using the draw providing the best trade-off based on our multi-criteria evaluation.

## Field implementation

Of the 295 PSUs sampled across all three jurisdictions, we sampled 85 PSUs in remote areas that we predicted would require charter flights to access. In Newfoundland, cost models accurately predicted an average cost of $1,721 (SD = $86) per PSU to access 16 remote PSUs, compared to actual costs which averaged $1,721 (SD = $441) per PSU (Fig 10). In comparison, the cost models for Saskatchewan predicted an average access cost of $3,741 (SD = $1,928) for the 43 remote PSUs sampled, but the actual access costs were substantially lower (mean = $1,573,

**Table 1. Linear model results comparing variation in access costs ($) between alternative monitoring designs.** The cost design was the reference category (Intercept).

| Design | β | SE | t-value |
|---|---|---|---|
| *Newfoundland* | | | |
| Intercept | 1,006,006.0 | 2870.2 | 350.51 |
| BOSS design | 121,535.7 | 3951.3 | 30.76 |
| Spatial design | 362,012.3 | 4072.3 | 88.90 |
| Habitat design | 482,229.8 | 4023.7 | 119.85 |
| *Saskatchewan* | | | |
| Intercept | 1,053,370.5 | 4927.6 | 213.77 |
| BOSS design | 175,634.7 | 7288.8 | 24.10 |
| Spatial design | 561,151.8 | 7587.2 | 73.96 |
| Habitat design | 1,020,042.6 | 8439.4 | 120.87 |
| *Yukon* | | | |
| Intercept | 331,627.1 | 990.8 | 334.70 |
| BOSS design | 8700.2 | 1615.1 | 5.39 |
| Spatial design | 541,671.3 | 1658.7 | 326.57 |
| Habitat design | 121,812.7 | 1507.5 | 80.80 |

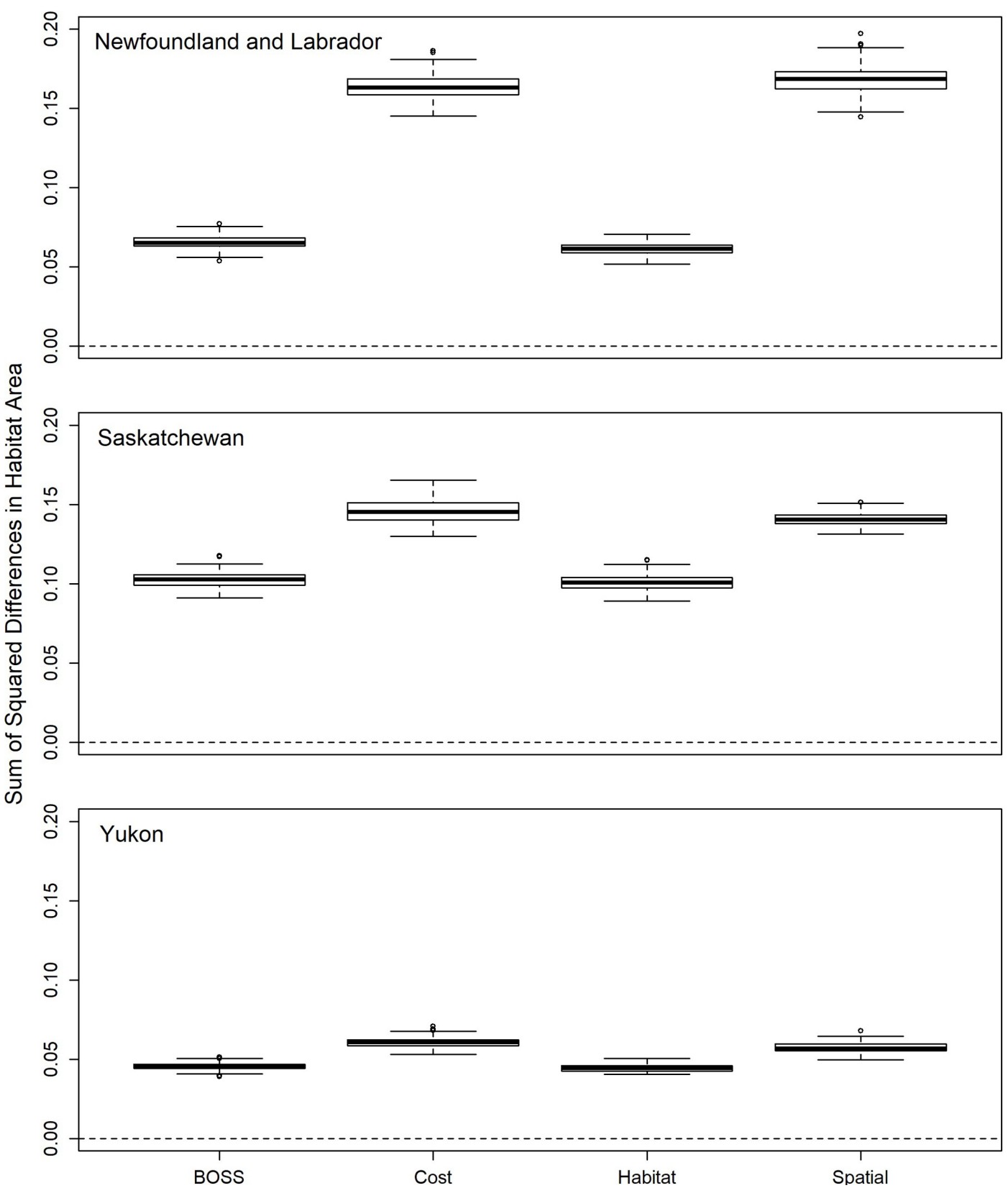

**Fig 7. Variation in habitat class representation (measured as the sum of square differences between the area of each habitat class within all PSUs in a randomized draw versus the proportional area of each habitat class, see Materials and Methods) between alternative sampling designs in Newfoundland and Labrador, Saskatchewan and Yukon Territory.** Data are from 100 separate random draws of 400 primary sampling units in each jurisdiction. Dark solid line represents the median, box indicates the inter-quartile range, whiskers are 1.5 times the inter-quartile range, and dots indicate extreme values.

SD = $1,562). In Yukon, the mean predicted access cost to reach the 26 remote PSUs sampled was $1,574 (SD = $82; Fig 10) per PSU, but the mean cost to access these PSUs was $3,016 (SD = $132). Weighted least squares regression (weights proportional to sample size) suggests across all three jurisdictions the access actual costs were 57% of the model predicted cost on average ($\beta = 0.57$, SE = 0.10; Fig 10).

## Discussion

We integrated key concepts from sampling theory to develop a hierarchically structured survey design that can provide stratified sampling of predefined strata (e.g., ecoregions) and achieve the desired spatial balance and habitat representation while minimizing access costs. The probability-based BOSS design allows for valid statistical inference from field samples across the sampling frame and reduces operational costs. We found that our integrated design was significantly less expensive than all but a design based only on access cost. Predicted access costs were 1.3–1.7 times more expensive for the habitat stratification design and 1.6–2.7 times more expensive on average for the spatial design compared to our BOSS design. Additionally, the distribution of habitat representation across sample draws from the BOSS design overlapped almost entirely with that from the habitat stratification design (see Fig 7). While we did find minor differences in the distribution of spatial balance between the designs considered (Fig 8), spatial balance was always close to a value of one, indicating the samples were well spread over jurisdictions irrespective of the type of design. Across jurisdictions, even the spatial design did not always produce spatial balance metrics that overlapped one, suggesting that the inclusion of unequal sampling probabilities in the BOSS design did not introduce systematic biases in spatial representation. Combined with our randomizations to balance trade-offs, we therefore

**Table 2. Linear model results comparing variation in habitat representation (i.e., sum of squared differences between area of land cover classes represented within sample draws against a balanced sample (see Methods)) between alternative sampling designs.** The habitat design was the reference category (Intercept).

| Design | β | SE | t-value |
|---|---:|---:|---:|
| *Newfoundland* | | | |
| Intercept | 0.06 | $3.84 \times 10^{-4}$ | 158.59 |
| BOSS design | $4.90 \times 10^{-3}$ | $5.60 \times 10^{-4}$ | 8.75 |
| Spatial design | 0.11 | $9.97 \times 10^{-4}$ | 108.35 |
| Cost design | 0.10 | $8.97 \times 10^{-4}$ | 114.37 |
| *Saskatchewan* | | | |
| Intercept | 0.04 | $2.06 \times 10^{-4}$ | 215.01 |
| BOSS design | $9.36 \times 10^{-4}$ | $3.10 \times 10^{-4}$ | 3.02 |
| Spatial design | 0.01 | $4.02 \times 10^{-4}$ | 32.31 |
| Cost design | 0.02 | $3.93 \times 10^{-4}$ | 41.56 |
| *Yukon* | | | |
| Intercept | 0.10 | $4.78 \times 10^{-4}$ | 211.13 |
| BOSS design | $1.53 \times 10^{-3}$ | $7.17 \times 10^{-4}$ | 2.14 |
| Spatial design | 0.04 | $6.47 \times 10^{-4}$ | 61.51 |
| Cost design | 0.04 | $8.40 \times 10^{-4}$ | 53.36 |

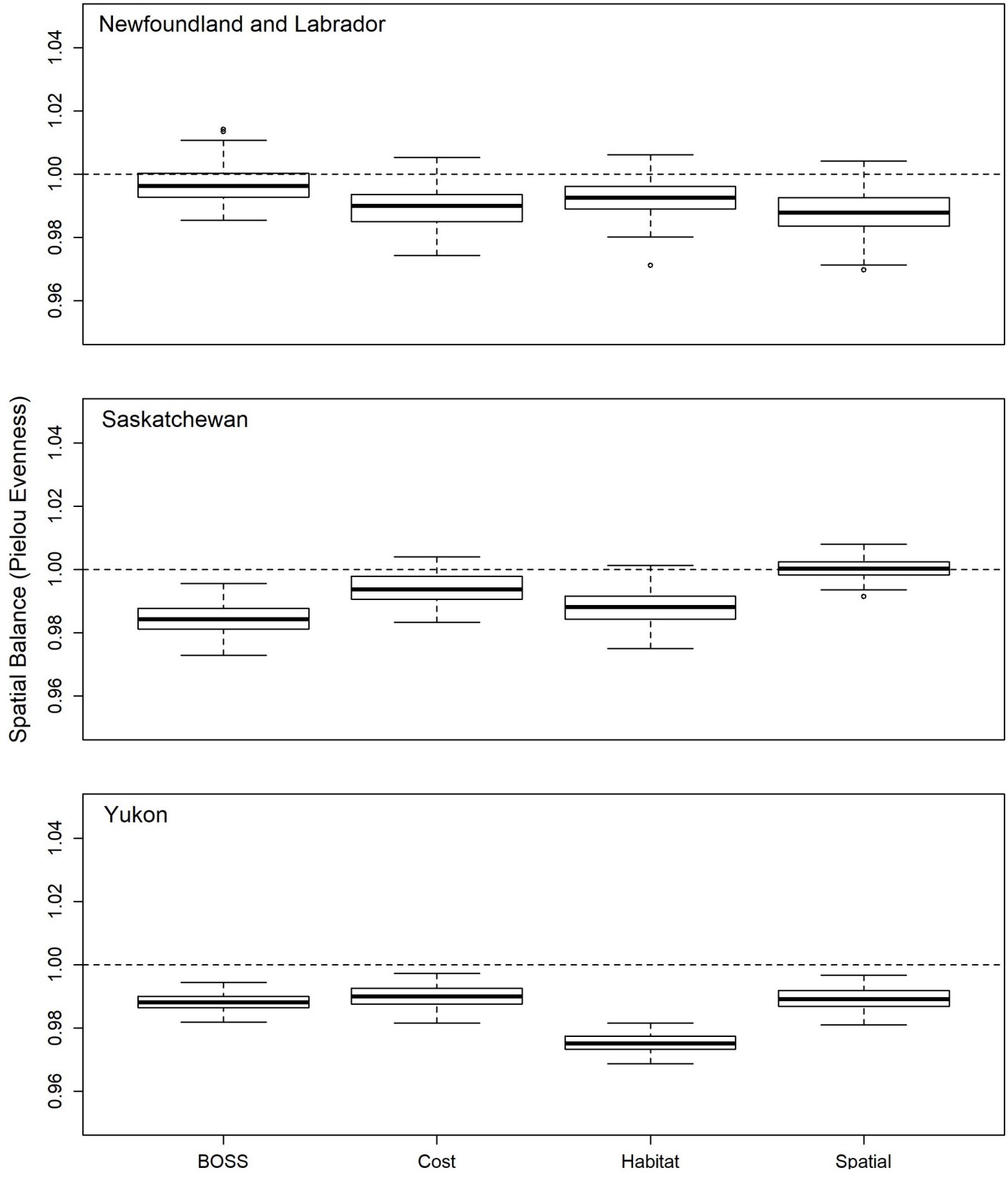

**Fig 8. Variation in spatial balance (Pielou evenness) between alternative sampling designs in Newfoundland, Saskatchewan and Yukon Territory.** Data are from 100 separate random draws of 400 primary sampling units in each jurisdiction. Dark solid line represents the median, box indicates the inter-quartile range, whiskers are 1.5 times the inter-quartile range, and dots indicate extreme values.

simultaneously achieved habitat and spatial representation goals while minimizing access costs.

Our pilot field seasons suggest that logistic considerations might result in access costs differing from those suggested by our cost models. While cost models for Newfoundland have thus far provided accurate prediction of average access costs per PSU, our cost models underestimated access costs in Yukon. Overall, regression analysis suggested that field implementation was 43% cheaper than suggested by our cost models. Many reasons likely contributed to the cost differences we observed, including external logistical support. One key difference is that our cost models largely portray the cost of access to a sample unit as independent of access to other nearby sampling units. In practice, we were able to combine logistics and access methods for many sampling units and thereby substantially lowered the average cost per sample unit. For example, in the summer of 2019 we chartered a Twin Otter to drop off two crews on or near opposite sides of Cree Lake, Saskatchewan (57°23′57″N 106°40′10″W) to access six PSUs via canoe. During crew retrieval, an additional crew was dropped off to canoe down river to a roadside pickup location, allowing access to two additional PSUs. We were therefore able to access eight PSUs for the cost of two charter flights (the base cost for accessing a single PSU), substantially reducing the average cost per PSU. We therefore recommend that programs revise their cost models as field implementation improves knowledge of local logistics. As a result, final program costs can be lowered by refinement of preliminary cost models.

Our sampling design has several key differences from other large-scale terrestrial bird monitoring programs that improve its utility for remote areas such as the boreal forest. Other programs such as the Integrated Monitoring in Bird Conservation Regions (IMBCR) program [9], the UK Breeding Bird Survey [60], the North American BBS [12] and the US National Park Service Vital Signs Monitoring program [61] also have defined sampling frames and randomly selected sample units that facilitate valid statistical inference from field samples [62].

**Table 3. Linear model results examining variation in spatial balance (Pielou evenness) between alternative sampling designs.** The equal-probability spatially balanced sampling design was the reference category (Intercept).

| Design | β | SE | t-value |
|---|---|---|---|
| *Newfoundland* | | | |
| Intercept | 0.99 | $6.52 \times 10^{-4}$ | 1514.78 |
| BOSS design | $8.70 \times 10^{-3}$ | $8.80 \times 10^{-4}$ | 9.87 |
| Cost design | $1.56 \times 10^{-3}$ | $9.03 \times 10^{-4}$ | 1.72 |
| Habitat design | $4.18 \times 10^{-3}$ | $8.92 \times 10^{-4}$ | 4.68 |
| *Saskatchewan* | | | |
| Intercept | 0.99 | $3.32 \times 10^{-4}$ | 2977.07 |
| BOSS design | $-1.10 \times 10^{-3}$ | $4.23 \times 10^{-4}$ | -2.61 |
| Cost design | $9.22 \times 10^{-4}$ | $4.52 \times 10^{-4}$ | 2.04 |
| Habitat design | $-1.39 \times 10^{-2}$ | $4.41 \times 10^{-4}$ | -31.61 |
| *Yukon* | | | |
| Intercept | 1.00 | $3.12 \times 10^{-4}$ | 3204.27 |
| BOSS design | $-1.59 \times 10^{-2}$ | $5.81 \times 10^{-4}$ | 27.41 |
| Cost design | $-6.15 \times 10^{-3}$ | $5.69 \times 10^{-4}$ | -10.82 |
| Habitat design | $-1.21 \times 10^{-2}$ | $5.89 \times 10^{-4}$ | -20.68 |

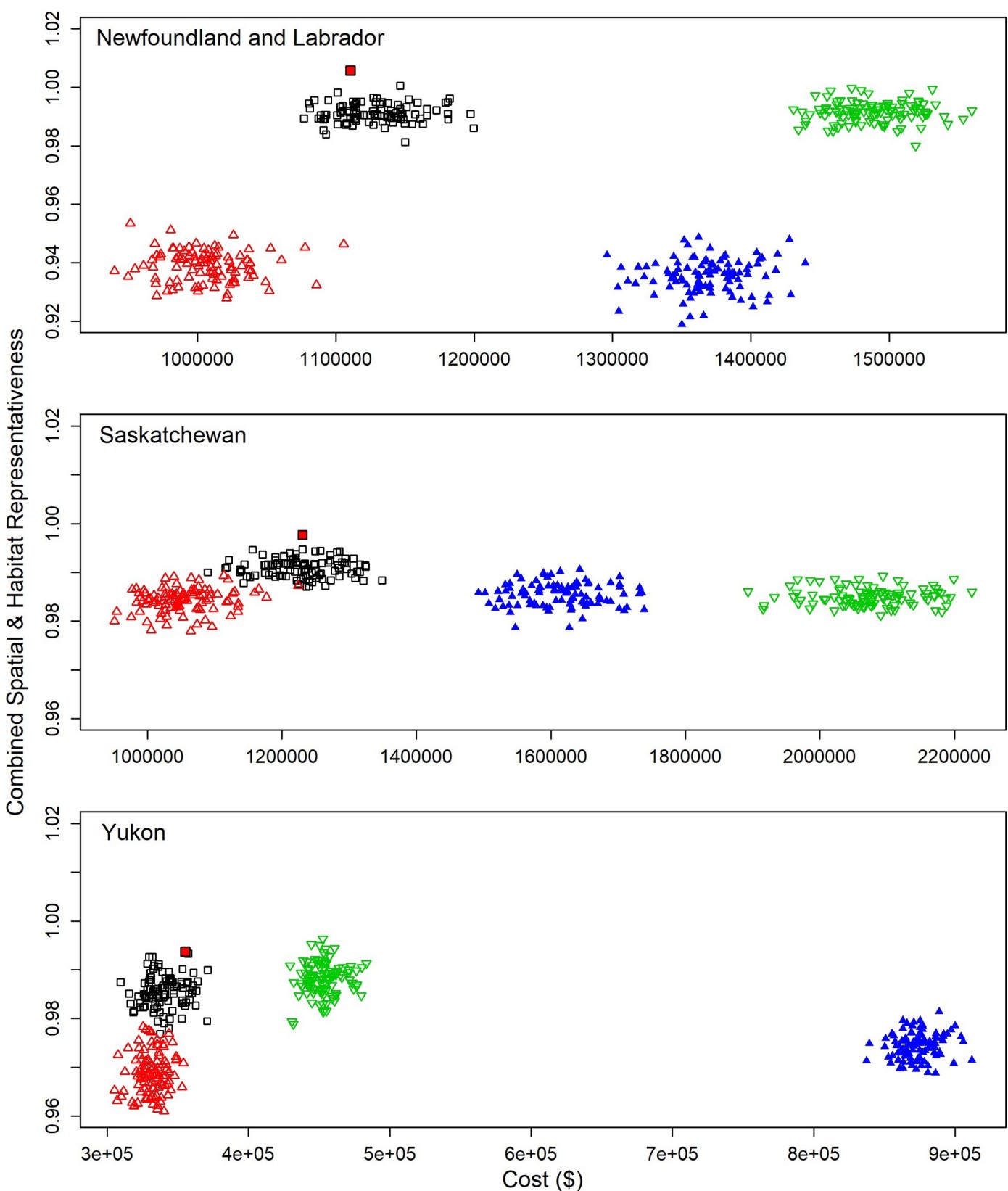

**Fig 9. Between draw variation in combined spatial and habitat (weighted sum, see Methods) representativeness of sampling relative to access costs in Newfoundland and Labrador, Saskatchewan and Yukon Territory for four alternative sampling designs: Cost based sampling design (open red triangles), the BOSS design (open black squares), the habitat design (open green triangles), and an equal probability spatial sampling (blue filled triangles).** The red-filled square within the cluster of points representing the BOSS design represents the lowest cost solution that maximizes combined spatial and habitat representation. Note differing x-axis scales due to variation in overall costs of operating in each jurisdiction.

Importantly, our design has fewer restrictions on the sampling frame while minimizing sampling costs, and implicitly accommodates health and safety considerations. While other programs restrict the sampling frame to roadsides or trail networks to facilitate access and alleviate safety concerns, we instead dealt with access by incorporating access costs into sample unit selection, and designing surveys to include multiple staff within primary sample units. Thus, our design increases safety and reduces costs without limiting statistical inference about bird populations to accessible areas [62–64]. In addition, our hierarchical stratification with static strata (BCRs, jurisdictions and ecoregions) allows for assessment of the population-level

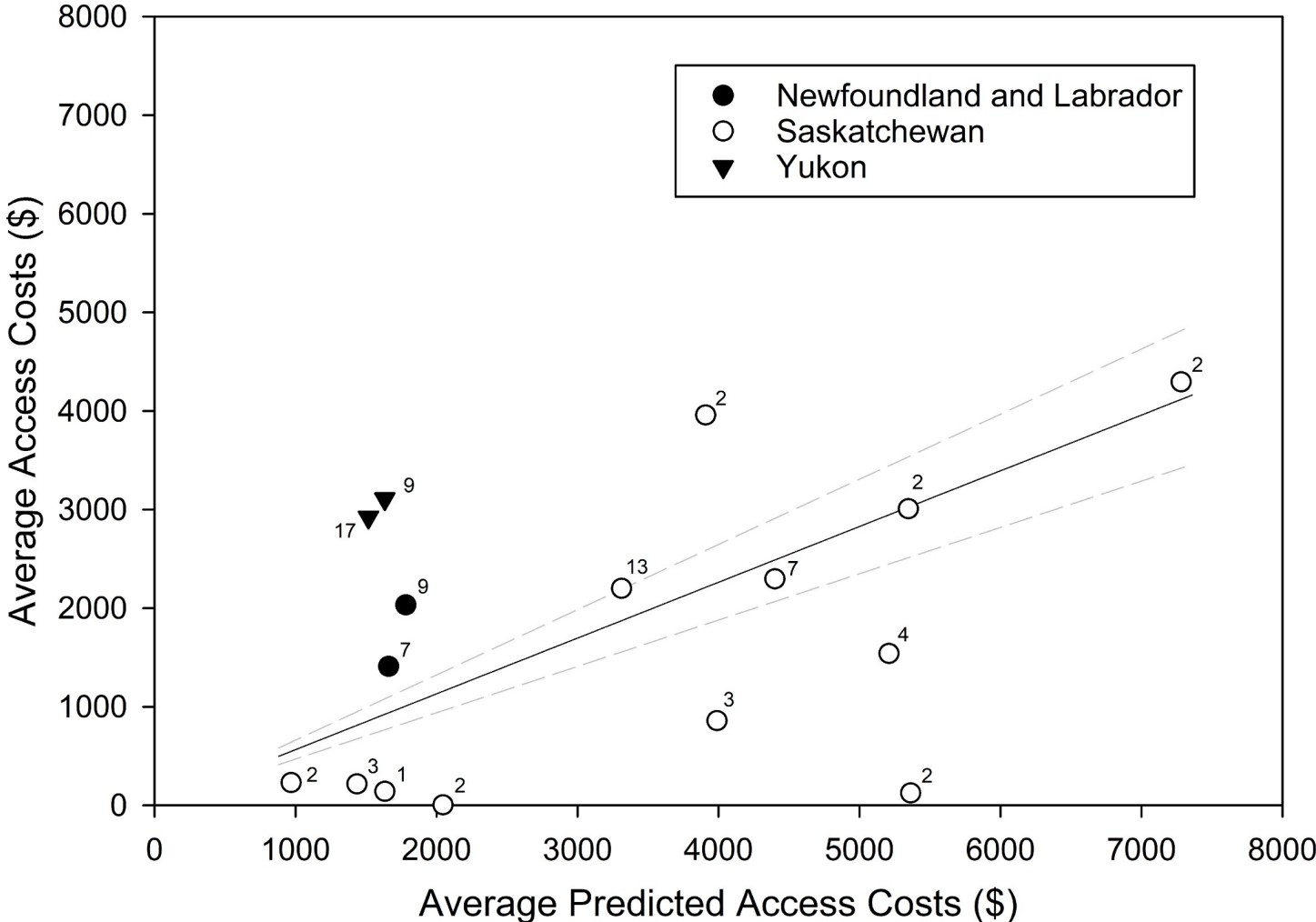

**Fig 10. Predicted versus realized access costs (in Canadian dollars) for each of Newfoundland and Labrador, Saskatchewan and Yukon Territory.** The solid line represents weighted (weights proportional to sample size) least squares fit and dashed lines represent 95% confidence intervals. Points represent average costs to access multiple (n = 1–17) primary sampling units, in line with how charter services were billed. Numbers beside symbols indicate the number of primary sampling units included with the associated mean cost estimate.

responses of birds to natural and human disturbance and climate change [9,65]. Indeed, Pavlacky et al. [9] recently demonstrated the ability of their hierarchical spatial design to allow estimation of both strata-specific population sizes and species-habitat relationships. Our approach of integrating both cost and weighted habitat-inclusion probabilities within a similar spatially balanced design will allow us to draw similar inferences to Pavlacky et al. [9] at multiple spatial scales, but at reduced costs compared to alternative approaches.

Our design directly incorporated several factors into the stratification and layout of primary and secondary sampling units to address the need to minimize costs and provide for the health and safety of field staff. Field implementation of the optimal designs selected here during pilot field seasons across all three jurisdictions showed that the BOSS design was achievable on the ground. We were able to apply the design across a wide variety of challenging conditions from mountainous terrain (e.g., >2000 m elevation), wetlands (floating bogs, fens and marshes), forests with blow-down (recent burns, tornados and insect outbreaks), as well as steep, rocky, and densely vegetated slopes across all three jurisdictions. Furthermore, drawing an oversample using the GRTS algorithm allowed us to replace unsafe PSUs without altering the representativeness of the sample [25]. Using pairs of observers to conduct surveys within each PSU means that staff can work alone to each survey a plot of nine SSUs within a PSU or they can work together to complete one plot of nine SSUs before travelling together to complete the second plot. The latter option would be appropriate if a PSU contains dangerous terrain, wildlife or limits to communication between staff (radio, satellite transmitter).

One complication imposed by our design is that most analyses assume that sampling is proportional to habitat availability and, by extension, population sizes. We used habitat stratification to ensure representation across the range of available habitat classes and thereby provide reasonable sample sizes to facilitate modeling abundance and distributions of species associated with rare habitats. Habitat stratification should also improve our future capacity to detect trends of species using rare habitats. In particular, our approach to habitat stratification should increase model precision for species using rare habitats at the cost of reducing precision for common species, a trade-off we feel is worthwhile because estimating trends, distribution and habitat relationships for rare species is always more limiting. However, extrapolating from the sample to estimate stratum-specific population sizes and/or population trends is more complicated because strata boundaries and sizes can change with shifts in vegetation through time [10]. This problem can be addressed analytically via post-hoc stratification [66], weighting via the inclusion probabilities [68,69], including random-effects for the strata indexed by the inclusion probabilities [70], or inclusion of interval-specific covariates reflecting habitat supply [69]. Recent spatially explicit approaches to avian population size estimation [71] provide one potential approach to build trend estimation models that could implicitly incorporate changes in strata sizes through time.

While the BOSS design presented here incorporates several key improvements over current large-scale bird monitoring designs, we foresee several areas where further improvements could occur. First, we stratified habitat using coarse satellite-derived products due to their availability and consistency across the entire sampling frame. Incorporating higher-resolution products capable of further stratifying habitat, and potentially better quantifying the amount and distribution of rare habitats, should improve our capacity to monitor species associated with these habitats. One possible approach worth investigation would be to create an avian habitat classification (*sensu* [45]) from higher-resolution data, such as forest inventory data, for the portion of the sample frame for which they may be available. Higher-resolution data could be integrated into the current framework by (1) using the coarse-resolution satellite data to select PSUs and using the high-resolution habitat classification to stratify SSUs within the selected PSUs, or (2) splitting the selection of PSUs between areas with versus without access to these high-resolution data products.

Second, once sufficient on-the-ground data are available, it would be desirable to re-examine the stratification by using the design-based sample to estimate spatial and temporal variance in bird community composition directly for optimal allocation, rather than the proxy variables used in the initial sample. We anticipate that sampling efficiency will improve by incorporating these direct estimates of variability into the design, but overall costs could be inflated if these data are not incorporated until they are available for all regions. Using data from preliminary rounds of sampling under the sampling design to validate allocation based on multivariate dispersion of proxy variables will ensure allocation between strata is cost effective, and could be used for second-phase sampling to improve the precision of estimates further [28].

Future simulation studies are needed to examine the likely trend precision that can be achieved under the BOSS design under alternative PSU revisit schedules (e.g., every 5, 10, or 20 years). Evaluating whether the BOSS emphasis on greater sampling in southern regions (Fig 5) could impede our ability to detect trends for northern-distributed species or potentially improve our capacity to detect anthropogenic impacts on avian populations due to the concentration of anthropogenic disturbance in southern regions would be an important addition to these simulation studies. Finally, optimizing the use and allocation of humans versus ARUs within our design is also a priority for future simulation modelling. Recent studies suggest that biases in species detection between human and ARU surveys can largely be controlled [72,73], but potential biases between single versus multiple visits would ideally be controlled by spatially balanced allocations of survey effort between humans and ARUs. However, logistical constraints and/or the aggregation of hard to detect species (e.g., secretive marsh birds) into a subset of habitat may make other approaches more efficient and is the subject of ongoing work.

To avoid inefficiencies, there is a clear need for improvements in cost models or habitat layers to be reflected in revised sample draws; however, revised sample draws should account for existing sampling under the BOSS design to avoid inducing spatial imbalance. Foster et al. [26] proposed using a squared-loss distance metric to alter the inclusion probabilities around "legacy" monitoring sites (i.e., pre-existing sampling locations that are known to be a randomly selected representative sample from a known sample frame [26]). This approach would provide a clear way to improve future sampling under the BOSS design without introducing biases and inefficiencies due to spatial imbalances. In addition to data from legacy sites, a large collection of historic data exist within the Boreal Avian Modelling (*hereafter* BAM) project database [74] for a large portion of boreal Canada. The difficulty with incorporating data from the BAM database is that these largely represent "iconic sites" (*sensu* [26]) which are either known to be non-randomly selected and/or there is insufficient documentation to know whether the sampling frame included the full suite of habitat classes. Including iconic sites in a randomized draw could introduce bias into the design and result in biased status and trend estimates that may not match those in the population [26]. Since these historic datasets could potentially add valuable ecological insights into factors affecting populations of boreal birds, it would be fruitful to consider when and how they could be included within a long-term monitoring design.

It is worth nothing that our approach to "optimizing" which randomized draw to use derives from the concept of Pareto optimality [34], but does not strictly meet these criteria. In optimization, a Pareto optimal solution is a solution in which the results cannot be modified without resulting in the objective function becoming worse. In our approach, this could only be accomplished by considering all permutations of a given sample size. Considering all permutations would become computationally difficult as well as result in a solution that is no longer truly random. As such, we feel our approach provides a reasonable approximation of optimality while maintaining the desirable feature of representing a randomized draw.

We have demonstrated that including access costs and habitat within a hierarchically structured sampling scheme was the most cost-efficient design that maintained the desired spatial

and habitat representation. Although we have focused on a design for monitoring boreal birds, our approach should be broadly applicable and adaptable as a template for other regions where remote locations form a large proportion of the sampling frame. We recommend that our approach be adapted for use in other large-scale studies such as Breeding Bird Atlases [75] if access and sampling costs are limiting for much, if not all, of the study areas. For example, the Saskatchewan Breeding Bird Atlas (https://sk.birdatlas.ca/) is currently using systematic sampling within accessible areas, and the BOSS design within the boreal portion of the province. To facilitate adaptation and implementation of our approach and/or further refinements to our method, we provide worked examples and scripts within the Supporting Information. Using design-based approaches to inform the design of large-scale studies would add to their statistical rigor and efficiency and facilitate their integration (i.e., treated as legacy sites) into long-term monitoring to provide more responsive conservation efforts in light of widespread declines in biodiversity [1–3]. Our hierarchical sampling design should be widely implemented to monitor boreal birds because it can provide an unbiased representation of when and where conservation and management should be targeted [2, 76]. Using monitoring data to inform conservation decisions is crucial, given ongoing and projected ecological changes in the boreal biome [18,37] and population declines already observed for boreal birds [3].

## Supporting information

**S1 Data. Data used as proxy variables for assumed spatial and temporal variation in boreal forest bird communities.**
(CSV)

**S2 Data. R script implementing primary and secondary sample unit selection, including an example calculation of access costs for Northwest Territories, Canada.**
(R)

**S3 Data. Allocation of 4980 primary sampling units amongst geographic strata (intersection between ecoregions and jurisdictions) across Canada.**
(CSV)

**S4 Data. zip file containing land cover data for habitat stratification derived from Latifovic et al.** (Latifovic, et al., 2012) used in association with R scripts in S2.
(ZIP)

**S5 Data. Shapefiles required to run example analyses in script S2.**
(ZIP)

**S6 Data. R script implementing parallel computing for primary and secondary sample unit selection for Manitoba, Canada, where access-cost models were created using ArcGIS using the Spatial Analyst™ extension.**
(R)

**S7 Data. Zip file containing example geospatial data, metadata describing of cost modeling, and supporting R scripts used in association with R scripts in S6.**
(ZIP)

**S8 Data. Comma-separated values file of data from 100 iterations of drawing 400 primary sample units for three Canadian jurisdictions (Newfoundland & Labrador, Saskatchewan and Yukon Territory).**
(CSV)

## Acknowledgments

We are extremely grateful to R. Weeber for discussions leading to several of the concepts incorporated within this design. We are grateful to Z. Li and K. Swiston for geographic information systems support. We thank M. Robertson, D. Iles and D. Hope for helpful comments on an earlier draft of this manuscript. F. Koch, C. Handel, and two anonymous reviewers provided helpful comments that improved the manuscript.

## Author Contributions

**Conceptualization:** Steven L. Van Wilgenburg, C. Lisa Mahon, Greg Campbell, Margaret Campbell, Wendy Easton, Charles M. Francis, Samuel Haché, Rhiannon F. Pankratz, Rich Russell, Adam C. Smith, Peter Thomas, Judith D. Toms, Junior A. Tremblay.

**Data curation:** Steven L. Van Wilgenburg, Greg Campbell, Logan McLeod.

**Formal analysis:** Steven L. Van Wilgenburg, Greg Campbell, Logan McLeod.

**Investigation:** Steven L. Van Wilgenburg, C. Lisa Mahon, Greg Campbell, Logan McLeod, Margaret Campbell, Samuel Haché, Rhiannon F. Pankratz.

**Methodology:** Steven L. Van Wilgenburg, C. Lisa Mahon, Greg Campbell, Logan McLeod, Margaret Campbell, Dean Evans, Caitlin Mader, Adam C. Smith, Judith D. Toms, Junior A. Tremblay.

**Project administration:** C. Lisa Mahon.

**Resources:** Margaret Campbell, Wendy Easton, Charles M. Francis, Samuel Haché, Craig S. Machtans, Peter Thomas.

**Software:** Steven L. Van Wilgenburg, Greg Campbell, Logan McLeod, Dean Evans, Caitlin Mader.

**Supervision:** Charles M. Francis, Craig S. Machtans, Peter Thomas.

**Validation:** Adam C. Smith, Judith D. Toms.

**Visualization:** Steven L. Van Wilgenburg, Logan McLeod, Dean Evans.

**Writing – original draft:** Steven L. Van Wilgenburg, C. Lisa Mahon, Logan McLeod.

**Writing – review & editing:** Steven L. Van Wilgenburg, C. Lisa Mahon, Greg Campbell, Wendy Easton, Charles M. Francis, Samuel Haché, Craig S. Machtans, Caitlin Mader, Rhiannon F. Pankratz, Rich Russell, Adam C. Smith, Peter Thomas, Judith D. Toms, Junior A. Tremblay.

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
