## [Decision Letter · Decision Letter 0]

6 Mar 2020

PONE-D-19-35579

A cost efficient spatially balanced hierarchical sampling design for monitoring boreal birds incorporating access costs and habitat stratification

PLOS ONE

Dear Mr Van Wilgenburg,

Thank you for submitting your manuscript to PLOS ONE. After careful consideration, we feel that it has merit but does not fully meet PLOS ONE’s publication criteria as it currently stands. Therefore, we invite you to submit a revised version of the manuscript that addresses the points raised during the review process.

**From the Academic Editor: I apologize for the delay in my initial decision, but I wanted to get comments from multiple reviewers regarding your manuscript, which covers a highly relevant topic that should be of interest to many PLOS ONE readers. The reviewers agree that the manuscript is very well written. Each reviewer has noted just a few aspects that could be improved. I ask that you consider their comments and revise your manuscript accordingly. I don't think this will require too much of your time, and should result in an even stronger paper.**

We would appreciate receiving your revised manuscript by Apr 20 2020 11:59PM. To enhance the reproducibility of your results, we recommend that if applicable you deposit your laboratory protocols in protocols.io, where a protocol can be assigned its own identifier (DOI) such that it can be cited independently in the future. For instructions see: http://journals.plos.org/plosone/s/submission-guidelines#loc-laboratory-protocols

We look forward to receiving your revised manuscript.

Kind regards,

Frank H. Koch, PhD

Academic Editor

PLOS ONE

Journal Requirements:

"This research was supported by Environment and Climate Change Canada."

4. We note that Figures #1-3 and 5 in your submission contain map images which may be copyrighted. All PLOS content is published under the Creative Commons Attribution License (CC BY 4.0), which means that the manuscript, images, and Supporting Information files will be freely available online, and any third party is permitted to access, download, copy, distribute, and use these materials in any way, even commercially, with proper attribution. For these reasons, we cannot publish previously copyrighted maps or satellite images created using proprietary data, such as Google software (Google Maps, Street View, and Earth). For more information, see our copyright guidelines: http://journals.plos.org/plosone/s/licenses-and-copyright.

1.    You may seek permission from the original copyright holder of Figures #1-3 and 5 to publish the content specifically under the CC BY 4.0 license. 

Reviewers' comments:

Reviewer's Responses to Questions

**Comments to the Author**

1. Is the manuscript technically sound, and do the data support the conclusions?

Reviewer #1: Yes

Reviewer #2: Yes

Reviewer #3: Yes

2. Has the statistical analysis been performed appropriately and rigorously? 

Reviewer #1: Yes

Reviewer #2: Yes

Reviewer #3: Yes

3. Have the authors made all data underlying the findings in their manuscript fully available?

Reviewer #1: Yes

Reviewer #2: Yes

Reviewer #3: Yes

4. Is the manuscript presented in an intelligible fashion and written in standard English?

Reviewer #1: Yes

Reviewer #2: Yes

Reviewer #3: Yes

5. Review Comments to the Author

Reviewer #1: Your well-written paper provides a superb blueprint for addressing a very important need, how best to monitor terrestrial breeding bird populations across the boreal region of Canada. You proposed four scientifically rigorous sampling designs and used simulation modeling to test their efficiencies relative to cost, habitat-representativeness, and spatial balance, all of which are important for implementing a successful and robust monitoring program. You also tested the sampling design empirically in three large study areas, comparing actual vs. projected costs for implementation, a pragmatic consideration for successful adoption and funding of such a large-scale program. Overall, your paper is based on sound analytical procedures, knowledge of how birds are distributed across the landscape, thoughtful reasoning about how populations are likely to change over time, and an understanding of the logistical features needed to implement such an ambitious program in remote areas. Including your R code further illustrates the complexity of the task you have undertaken and will be extremely helpful to those who want to understand the details about how best to design such a program.

There are a few aspects that you could address more fully. First, you use maximization of the variance of diversity of species (with habitat as proxy) rather than minimization of the variance in estimates of population trends or population size as one of the key criteria for allocating effort in terms of primary sampling units (PSUs). I realize that optimization relative to estimating population size or trend would be difficult to achieve, particularly for an omnibus survey that encompasses a multitude of species with varying levels of population change and abundance. I surmise, however, that the proposed allocation would tend to weight southern areas of the boreal more heavily (where species richness is higher) and that there may be low power to estimate trends for populations that are declining rapidly (and have lower population sizes) or those that may be changing rapidly in northern ecotones (perhaps even increasing). It would be interesting if you could address some of these limitations or tradeoffs in the discussion.

Secondly, you mention that the monitoring program will use point count surveys with trained observers and/or autonomous recording units (ARUs). I assume that the decision on which of these methods to use would be related to cost of access for any given PSU. If so, I would conclude that more remote, northern areas could conceivably be monitored exclusively through the use of ARUs alone, because they can be deployed and retrieved least expensively via snow machine or ice roads in winter. It would be useful if the authors could expound at least minimally upon some of the implications of this aspect of the design. Use of ARUs alone without validation from comparable surveys by human observers can result in biased estimates of species’ occurrence and density, particularly for those species that are detected more often by visual than aural cues. ARUs can provide more detailed information on temporal and spatial occurrence of species in a given area if they are set to record samples throughout the season. Analysis of recordings, however, incurs a significant cost, particularly for monitoring of all species, which at this point cannot be automated efficiently. Also, I don’t believe that the cost of the ARUs was factored into the cost of access, but perhaps it should be. In the end, it might be worth a point of discussion to address some of the tradeoffs of using a combination of human observers and ARUs and to suggest that this is one other aspect of efficiency that remains to be designed. I would advocate for a spatially balanced use of human observers and ARUs so that any habitat- or temporal-related biases can be estimated and corrected for. A couple of other recent papers on ARUs in sparse northern habitats you might examine include Thompson et al. (2017, J. Wildl. Manage. 81:1228-1241) and Vold et al. (2017, Wildl. Soc. Bull. 10.1002/wsb.785).

Thirdly, I was a bit confused about how secondary sampling units (SSUs) were selected relative to the presence of water. It seems that any PSUs that encompassed only water were excluded from the sampling frame, which is reasonable. However, it was unclear how the mini-grids of 3 x 3 points were selected with respect to water features within a given PSU. From a cursory look at the R code, it appears that all points that fell within water were also removed from the sampling frame. I’m not certain what this did in terms of influencing which grids might be selected, but you might want to ensure that you are not biasing (low) your sample of points adjacent to water bodies, which are extremely important in terms of species richness and density in the boreal region.

Fourthly, given your proposed, very complicated sampling design, I anticipate that statistical analyses will be extremely complex, which you acknowledge in the discussion. The complexity will depend, of course, on what parameter you are estimating (e.g., habitat relationships, density, changes in distribution, or population trends). Once you add in the temporal dimension, with replication across years, the complexity will be even more challenging, particularly if your sampling intensity or inclusion probabilities change through time. I caution against changing the boundaries of your strata through time for long-term monitoring of population trends, particularly relative to habitat. Not only will vegetation be changing through time, but community composition will likely shift as well, with individual species responding differentially to changes in vegetation, temperature, precipitation, predator abundance, and other ecological factors. Thus, if you retain habitat diversity as a major determinant of inclusion probability, you should consider the ramifications of altering that through time. There are many powerful techniques for modeling trends through time, so this might not be a problem. There could be consequences, however, for the resulting precision of your estimates.

Pertinent to this point is the temporal replication of sampling. It would be helpful to mention what you have in mind for this aspect of your sampling design, particularly for estimating changes in distribution or population size (will there be annual estimates of population trend?). You should also address the possible ramifications of missing data from repeated samples, which will inevitably occur and will prove no small headache, particularly in remote areas where access is so challenging and unpredictable. Typically, repeated samples at the same sites will be more efficient statistically in estimating population trends, but an alternative would be to set up different sampling frames at various intervals through time, with changes in inclusion probabilities. In the boreal zone, sampling individual sites on a biennial basis may be more efficient than sampling them every year because of high interannual correlation (Handel and Sauer 2017). I was glad to see you consider the important issue of how best to incorporate legacy sites. Such data sets can provide key information on long-term changes in distribution and abundance but provide their own challenges when trying to account for potential bias in terms of selection of samples.

Finally, I note a few other minor points that would be helpful to clarify or correct:

P. 11, line 8. What landcover map did you use and how many cover classes were there?

P. 15, line 16. Ralph et al. (1993) is missing from Lit Cited and Matsuoka et al. (2014) should be cited as numbered reference. I also noted that references 71-80 don’t seem to be cited in the text.

P. 19, line 14. Final sentence of the paragraph seems to be missing text after ‘where.’

P. 21, line 17. I would change this to ‘red-filled squares’ so that it is clear what you are referencing. It took me a bit to figure this out (I had to go back to methods to understand what you were saying, especially since red is also used for the triangles representing cost).

P. 22, lines 6-10. I think you meant to reference Fig. 10 instead of Fig. 8. I was confused, however, by the number of PSUs listed. I counted only 2 PSUs in Newfoundland and Labrador (not 16), 2 in the Yukon (not 26), and 12 in Saskatchewan (not 43).

P. 23, lines 9-12. I did not understand the sentence beginning “In addition, the distribution of spatial balance metrics…”

P. 28, line 6. Two l’s in Boreal Avian ‘Modelling.’

Figure 1. You might consider outlining jurisdictions that you sampled in bold lines.

Figure 2. Consider adding a distance scale to each of these submaps or else noting in the figure heading that hexagons are 5 km in diameter and points are spaced 300 m apart.

Figure 4. ‘Proportional’ is misspelled on x-axis label.

I commend you all for this impressive and well-executed effort.

Colleen M. Handel

USGS Alaska Science Center

Reviewer #2: This is a nicely written paper. I would suggest that the authors provide a stronger background on spatial sampling, and sampling optimization:

Delmelle, E. M., & Goovaerts, P. (2009). Second-phase sampling designs for non-stationary spatial variables. Geoderma, 153(1-2), 205-216.

Van Groenigen, J. W., Stein, A., & Zuurbier, R. (1997). Optimization of environmental sampling using interactive GIS. Soil Technology, 10(2), 83-97.

Delmelle, E. (2009). Spatial sampling. The SAGE handbook of spatial analysis, 183, 206.

Reviewer #3: This paper by Van Wilgenburg et al. is well-written, technically sound, and of broad interest to the research community. There are too few papers on the topic of sampling design when it comes to broad scale monitoring programs supported by public funds; and it is critical that we develop strategic methods in areas like the Boreal where many countries have an international responsibility to maintain biodiversity in the face of multiple stressors.

I selected minor revisions because most of my recommended changes can be made fairly easily and no re-analysis or major reworking of the MS are necessary. I made comments directly on the pdf if that helps. More substantive comments can be found from p12 on. My most major concern is that the authors need to carefully consider their use of the term optimal. The optimization literature uses this term in a very specific and mathematical way and it is not clear from this work that the authors performed an optimization. I think a better way of describing what they did was to perform a spatial benefits costs analysis of various broad scale monitoring strategies. They do balance trade offs, but it is not clear that their preferred design is 'optimal' per se. I also think that they might want to call their approach something other than Boreal Optimal Sampling Strategy (BOSS).

The discussion is on the long side. It talks a lot about the advantages of their approach. I think the authors can shorten it. But I also think they should work in a discussion of other literature that examines large scale monitoring programs and trade offs between spatial coverage, costs, and target species representation.

6. PLOS authors have the option to publish the peer review history of their article (what does this mean?). If published, this will include your full peer review and any attached files.

Reviewer #1: Yes: Colleen M Handel

Reviewer #2: No

Reviewer #3: No

---

## [Author Response · Author response to Decision Letter 0]

23 Apr 2020

We would like to thank the reviewers for a very collegial and constructive set of reviews. You will find our responses to your queries below. We hope that you find our responses have adequately answered your questions.

Comments from Reviewer 1:

Reviewer #1: Your well-written paper provides a superb blueprint for addressing a very important need, how best to monitor terrestrial breeding bird populations across the boreal region of Canada. You proposed four scientifically rigorous sampling designs and used simulation modeling to test their efficiencies relative to cost, habitat-representativeness, and spatial balance, all of which are important for implementing a successful and robust monitoring program. You also tested the sampling design empirically in three large study areas, comparing actual vs. projected costs for implementation, a pragmatic consideration for successful adoption and funding of such a large-scale program. Overall, your paper is based on sound analytical procedures, knowledge of how birds are distributed across the landscape, thoughtful reasoning about how populations are likely to change over time, and an understanding of the logistical features needed to implement such an ambitious program in remote areas. Including your R code further illustrates the complexity of the task you have undertaken and will be extremely helpful to those who want to understand the details about how best to design such a program.

There are a few aspects that you could address more fully. First, you use maximization of the variance of diversity of species (with habitat as proxy) rather than minimization of the variance in estimates of population trends or population size as one of the key criteria for allocating effort in terms of primary sampling units (PSUs). I realize that optimization relative to estimating population size or trend would be difficult to achieve, particularly for an omnibus survey that encompasses a multitude of species with varying levels of population change and abundance. I surmise, however, that the proposed allocation would tend to weight southern areas of the boreal more heavily (where species richness is higher) and that there may be low power to estimate trends for populations that are declining rapidly (and have lower population sizes) or those that may be changing rapidly in northern ecotones (perhaps even increasing). It would be interesting if you could address some of these limitations or tradeoffs in the discussion.

Response: Thank you very much, this highlights a need for us to further elaborate and should therefore be a substantial improvement since this is a key component of our approach. You are correct that our stratification will emphasize increased sampling effort in regions with the greatest variance. It is important to note however that optimal sampling theory suggests that increasing sampling in strata with the greatest variance will result in increased precision of the estimate(s) of annual abundance, and therefore trend. We now elaborate on this in the text (1st paragraph of the “Sample size allocation” subsection of the Methods). The addition of a weighting for avian species richness does tend to emphasize sampling in southern regions (Figure 5); however, we see this as advantageous owing to the number of species involved, the access costs, and the spatial overlap with anthropogenic threats. It is true that low densities of northern distributed species may magnify this issue, but an initial sample collected under our design will help to further evaluate the trade-offs involved (e.g. in future simulation modeling). We elaborate on this in the discussion (8th paragraph), and now further discuss how we will re-evaluate this stratification using bird data collected under our design i.e. instead of environmental proxies.

Secondly, you mention that the monitoring program will use point count surveys with trained observers and/or autonomous recording units (ARUs). I assume that the decision on which of these methods to use would be related to cost of access for any given PSU. If so, I would conclude that more remote, northern areas could conceivably be monitored exclusively through the use of ARUs alone, because they can be deployed and retrieved least expensively via snow machine or ice roads in winter. It would be useful if the authors could expound at least minimally upon some of the implications of this aspect of the design. Use of ARUs alone without validation from comparable surveys by human observers can result in biased estimates of species’ occurrence and density, particularly for those species that are detected more often by visual than aural cues. ARUs can provide more detailed information on temporal and spatial occurrence of species in a given area if they are set to record samples throughout the season. Analysis of recordings, however, incurs a significant cost, particularly for monitoring of all species, which at this point cannot be automated efficiently. Also, I don’t believe that the cost of the ARUs was factored into the cost of access, but perhaps it should be. In the end, it might be worth a point of discussion to address some of the tradeoffs of using a combination of human observers and ARUs and to suggest that this is one other aspect of efficiency that remains to be designed. I would advocate for a spatially balanced use of human observers and ARUs so that any habitat- or temporal-related biases can be estimated and corrected for. A couple of other recent papers on ARUs in sparse northern habitats you might examine include Thompson et al. (2017, J. Wildl. Manage. 81:1228-1241) and Vold et al. (2017, Wildl. Soc. Bull. 10.1002/wsb.785).

Response: Here we have chosen to focus on the implications of alternative approaches to stratification and selection of the spatial locations for sampling, but see the points you raise here as key next steps in our program development. Decisions on where and when to allocate ARUs can vary substantially based on access logistics, potential partnerships, etc. that add further complications to the decision. We agree with your suggestion of spatially balanced use ARUs versus human observations would be the optimal approach to avoid spatial, habitat and temporal biases. We briefly (paragraph 8) discuss this in the discussion, but largely point to ongoing and future work that will help us optimize this component of our program.

Thirdly, I was a bit confused about how secondary sampling units (SSUs) were selected relative to the presence of water. It seems that any PSUs that encompassed only water were excluded from the sampling frame, which is reasonable. However, it was unclear how the mini-grids of 3 x 3 points were selected with respect to water features within a given PSU. From a cursory look at the R code, it appears that all points that fell within water were also removed from the sampling frame. I’m not certain what this did in terms of influencing which grids might be selected, but you might want to ensure that you are not biasing (low) your sample of points adjacent to water bodies, which are extremely important in terms of species richness and density in the boreal region.

Response: You are correct that we did indeed eliminate grid points that fell in open water. We also set a minimum target of points to ensure we can meet our within PSU sampling goals. This along with the systematic nature of the grids that are created by the algorithm tends to ensure sampling across a gradient of distances from water (even if the centroid does not fall in or immediately adjacent to water). For example, samples visited under this design thus far in Saskatchewan have represented a range of 0m – 5791 m from large waterbodies, with 5% of the sampling occurring less than 50 m from a waterbody, 12.5% less than 124m, 25% within 218m of water. Below are quantiles of that distribution.

0% 5% 12.5% 25% 50% 75% 87.5% 95% 100%

0m 46m 124m 218m 457m 819m 1231m 1788m 5791m

Fourthly, given your proposed, very complicated sampling design, I anticipate that statistical analyses will be extremely complex, which you acknowledge in the discussion. The complexity will depend, of course, on what parameter you are estimating (e.g., habitat relationships, density, changes in distribution, or population trends). Once you add in the temporal dimension, with replication across years, the complexity will be even more challenging, particularly if your sampling intensity or inclusion probabilities change through time. I caution against changing the boundaries of your strata through time for long-term monitoring of population trends, particularly relative to habitat. Not only will vegetation be changing through time, but community composition will likely shift as well, with individual species responding differentially to changes in vegetation, temperature, precipitation, predator abundance, and other ecological factors. Thus, if you retain habitat diversity as a major determinant of inclusion probability, you should consider the ramifications of altering that through time. There are many powerful techniques for modeling trends through time, so this might not be a problem. There could be consequences, however, for the resulting precision of your estimates.

RESPONSE: We appreciate that the analyses will be more complicated by our choice of design. We have however considered the implications for resulting estimates of trend precision and we anticipate that our choices will result in decreased precision for common species, but should improve precision and accuracy for species associated with rare habitats. As such, we feel that this trade-off will be worthwhile because it is the rare species for which we always have the greatest difficulty estimating trends. We have added brief text to the discussion (paragraph 5) to address this.

Pertinent to this point is the temporal replication of sampling. It would be helpful to mention what you have in mind for this aspect of your sampling design, particularly for estimating changes in distribution or population size (will there be annual estimates of population trend?). You should also address the possible ramifications of missing data from repeated samples, which will inevitably occur and will prove no small headache, particularly in remote areas where access is so challenging and unpredictable. Typically, repeated samples at the same sites will be more efficient statistically in estimating population trends, but an alternative would be to set up different sampling frames at various intervals through time, with changes in inclusion probabilities. In the boreal zone, sampling individual sites on a biennial basis may be more efficient than sampling them every year because of high interannual correlation (Handel and Sauer 2017). I was glad to see you consider the important issue of how best to incorporate legacy sites. Such data sets can provide key information on long-term changes in distribution and abundance but provide their own challenges when trying to account for potential bias in terms of selection of samples.

RESPONSE: We have added brief text to the discussion detailing our longer-term plans. In brief, we are currently focusing on using our design to collect data to fill key gaps in our knowledge on abundance and distribution. These data will subsequently inform simulation analyses to compare and contrast alternative rotating panel designs. We envision revisiting the majority of sites on a five or ten year revisit schedule and have a subset of sites with greater temporal replication (annually) to allow improve precision of trend estimates and estimation of inter-annual variance. Pragmatically, this will involve cost benefit trade-offs and arguments for further resourcing, and thus will not be immediately settled. We attempt to briefly outline future work that will address these issues in the discussion (paragraph 8).

Finally, I note a few other minor points that would be helpful to clarify or correct:

P. 11, line 8. What landcover map did you use and how many cover classes were there?

RESPONSE: We have now added reference to the data source here and discuss the number of landcover classes involved

P. 15, line 16. Ralph et al. (1993) is missing from Lit Cited and Matsuoka et al. (2014) should be cited as numbered reference. I also noted that references 71-80 don’t seem to be cited in the text.

RESPONSE: Added reference to Matsuoka et al and Ralph et al manuscripts and have removed the other references which were included by accident from the reference management software. Thanks for catching this.

P. 19, line 14. Final sentence of the paragraph seems to be missing text after ‘where.’

RESPONSE: Deleted ‘where’.

P. 21, line 17. I would change this to ‘red-filled squares’ so that it is clear what you are referencing. It took me a bit to figure this out (I had to go back to methods to understand what you were saying, especially since red is also used for the triangles representing cost).

RESPONSE: Done.

P. 22, lines 6-10. I think you meant to reference Fig. 10 instead of Fig. 8. I was confused, however, by the number of PSUs listed. I counted only 2 PSUs in Newfoundland and Labrador (not 16), 2 in the Yukon (not 26), and 12 in Saskatchewan (not 43).

RESPONSE: We have corrected the figure reference in the text. We understand the confusion regarding sample sizes. This stems from the figure depicting average access costs to multiple PSUs within the same air charter contract(s) since billing is done on a contract by contract basis and not PSU by PSU. Thus, a single data point may represent access to many PSUs. For example, one of the Yukon data points represents access of 17 PSUs while the data point represents access of 9 PSUs. We have added text to the figure heading to clarify this. 

P. 23, lines 9-12. I did not understand the sentence beginning “In addition, the distribution of spatial balance metrics…”

RESPONSE: Reworded to “Across jurisdictions, spatial balance metrics of the spatial design did not always overlap one; suggesting that the inclusion of unequal sampling probabilities did not introduce systematic biases in spatial representation”

P. 28, line 6. Two l’s in Boreal Avian ‘Modelling.’

RESPONSE: Revised

Figure 1. You might consider outlining jurisdictions that you sampled in bold lines.

RESPONSE: We have added the outline as suggested 

Figure 2. Consider adding a distance scale to each of these submaps or else noting in the figure heading that hexagons are 5 km in diameter and points are spaced 300 m apart.

RESPONSE: Distance scale added as requested

Figure 4. ‘Proportional’ is misspelled on x-axis label.

RESPONSE: Corrected

Comments from Reviewer #2: This is a nicely written paper. I would suggest that the authors provide a stronger background on spatial sampling, and sampling optimization:

Delmelle, E. M., & Goovaerts, P. (2009). Second-phase sampling designs for non-stationary spatial variables. Geoderma, 153(1-2), 205-216.

Van Groenigen, J. W., Stein, A., & Zuurbier, R. (1997). Optimization of environmental sampling using interactive GIS. Soil Technology, 10(2), 83-97.

Delmelle, E. (2009). Spatial sampling. The SAGE handbook of spatial analysis, 183, 206.

RESPONSE: Thank you for theses helpful references. We have used these and added text providing slightly more reference to and explanation of spatial sampling in the second last paragraph of the introduction and paragraph seven of the discussion.

Comments from Reviewer #3: This paper by Van Wilgenburg et al. is well-written, technically sound, and of broad interest to the research community. There are too few papers on the topic of sampling design when it comes to broad scale monitoring programs supported by public funds; and it is critical that we develop strategic methods in areas like the Boreal where many countries have an international responsibility to maintain biodiversity in the face of multiple stressors.

I selected minor revisions because most of my recommended changes can be made fairly easily and no re-analysis or major reworking of the MS are necessary. I made comments directly on the pdf if that helps. More substantive comments can be found from p12 on. My most major concern is that the authors need to carefully consider their use of the term optimal. The optimization literature uses this term in a very specific and mathematical way and it is not clear from this work that the authors performed an optimization. I think a better way of describing what they did was to perform a spatial benefits costs analysis of various broad scale monitoring strategies. They do balance trade offs, but it is not clear that their preferred design is 'optimal' per se. I also think that they might want to call their approach something other than Boreal Optimal Sampling Strategy (BOSS).

RESPONSE: Thank you for the comments and we can fully appreciate the confusion here. Our use of the term optimal actually derived from the sampling theory literature and not the optimization literature, but the confusion is understandable given our further attempts apply concepts from the optimization literature to select from amongst multiple randomized draws. We now clarify that our use of the term optimal refers to optimal allocation as per the sampling theory literature, and have reworded text around our use of multiple randomized draws to avoid confusion. We further cite the optimization literature related to the weighted sum approach we used in combining the multiple objectives we are attempting to maximize while minimizing costs. In addition, we have added brief text in the discussing Pareto optimality and discussing the pros and cons of our approach to “optimizing” the design.

We also incorporated most of the suggestions made directly within the manuscript. We only provide a detailed response to the more complicated questions/comments or ones where we have chosen not to incorporate below:

In regards to Equation 3, we have modified the equation to note that all pixels (1…j) within the primary sampling unit.

With regard to the note on the 1:1 correspondence line (in the Sample Size Allocation subsection of Results), we have not highlighted accuracy as mentioned, but add further emphasis on values above the 1:1 correspondence line later in the paragraph to highlight increased sampling of more variable strata.

With regard to the comment in the discussion re: “This could be modelled spatially using a more complicated spatial optimization algorithms that factor in reduced costs with neighbouring samples. (e.g., spatial simulated annealing algorithms)”. We have not added text in this regard as we do not yet feel we can sufficiently model these as yet hard to predict logistical efficiencies, though we hope to use our increasing experience on the ground to build improved models in the future. We envision using these refined models to improve our cost models within the same framework to maintain randomized sampling rather than using simulated annealing to seek a global optimum since environmental conditions in the boreal forest change rapidly and thus a global optimum may be a fleeting concept.

With respect to the comment: “Discussion is a bit on the long side and it focuses a lot on advantages of BOSS method. Can it be shortened, but also include reference to other large scale monitoring program designs that consider cost, coverage, and strata? ”. We have endeavored to shorten the text where we could. We note however that many of the requested revisions made by reviewer one have resulted in a slightly longer discussion. We feel that the remaining text is relevant to our results and thus would suggest it remain. We would however welcome suggestions from the Academic editor as to whether there remain any sections that they feel are tangential to the main thrust of the MS. With respect to comparisons against other designs, we are unaware of any that have incorporated all the concepts we have applied, and have kept comparisons to other progams within paragraph 3 of the discussion.

---

## [Editor Report · Decision Letter 1]

29 Apr 2020

PONE-D-19-35579R1

A cost efficient spatially balanced hierarchical sampling design for monitoring boreal birds incorporating access costs and habitat stratification

PLOS ONE

Dear Mr Van Wilgenburg,

Thank you for submitting your manuscript to PLOS ONE. After careful consideration, we feel that it has merit but does not fully meet PLOS ONE’s publication criteria as it currently stands. Therefore, we invite you to submit a revised version of the manuscript that addresses the points raised during the review process.

I appreciate the time and care you put into your responses to the reviewers. I believe you have addressed their concerns fully, and the manuscript is nearly suitable for publication. In reading through revision 1, I noticed some minor editorial errors (omitted words, punctuation, or the like). Rather than list them here, I've attached a tracked changes version of the manuscript for your reference. Once you address these and resubmit, I will move quickly to accept your manuscript. You do not need to submit point-by-point responses to any of my edits or comments.

We would appreciate receiving your revised manuscript by Jun 13 2020 11:59PM. To enhance the reproducibility of your results, we recommend that if applicable you deposit your laboratory protocols in protocols.io, where a protocol can be assigned its own identifier (DOI) such that it can be cited independently in the future. For instructions see: http://journals.plos.org/plosone/s/submission-guidelines#loc-laboratory-protocols

A marked-up copy of your manuscript that highlights changes made to the original (R1) version. This file should be uploaded as separate file and labeled 'Revised Manuscript with Track Changes'.An unmarked version of your revised paper without tracked changes. This file should be uploaded as separate file and labeled 'Manuscript'.

We look forward to receiving your revised manuscript.

Kind regards,

Frank H. Koch, PhD

Academic Editor

PLOS ONE
---

## [Author Response · Author response to Decision Letter 1]

22 May 2020

All changes made by the Academic Editor within the Manuscript Word file have been accepted. We made some additional changes to ensure clarity, accuracy, and adherence to the submission guidelines. We hope everything is now suitably formatted for publication.

---

## [Editor Report · Decision Letter 2]

28 May 2020

A cost efficient spatially balanced hierarchical sampling design for monitoring boreal birds incorporating access costs and habitat stratification

PONE-D-19-35579R2

Dear Dr. Van Wilgenburg,

We are pleased to inform you that your manuscript has been judged scientifically suitable for publication and will be formally accepted for publication once it complies with all outstanding technical requirements.

With kind regards,

Frank H. Koch, PhD

Academic Editor

PLOS ONE

Additional Editor Comments (optional):

Thank you for addressing the latest set of comments. The manuscript reads well and should interest many in the PLOS ONE audience.
---

## [Editor Report · Acceptance letter]

1 Jun 2020

PONE-D-19-35579R2 

A cost efficient spatially balanced hierarchical sampling design for monitoring boreal birds incorporating access costs and habitat stratification 

Dear Dr. Van Wilgenburg:

I am pleased to inform you that your manuscript has been deemed suitable for publication in PLOS ONE. Congratulations! Your manuscript is now with our production department. 

With kind regards,

on behalf of

Dr. Frank H. Koch 

Academic Editor

PLOS ONE